# Nuclear retention of unspliced HIV-1 RNA as a reversible post-transcriptional block in latency

Agnieszka Dorman[1,2,13], Maryam Bendoumou[3,13], Aurelija Valaitienė[4], Jakub Wadas [1,2], Haider Ali [1,2], Antoine Dutilleul[3], Paolo Maiuri [5], Lorena Nestola[3], Monika Bociaga-Jasik[6], Gilbert Mchantaf [7,8,9], Coca Necsoi[10], Stéphane De Wit[10], Véronique Avettand-Fenoël[7,8,9], Alessandro Marcello [11], Krzysztof Pyrc [12] ✉, Alexander O. Pasternak [4,14] ✉, Carine Van Lint [3,14] ✉ & Anna Kula-Pacurar [1,14] ✉

HIV-1 latency is mainly characterized at transcriptional level, and little is known about post-transcriptional mechanisms and their contribution to reactivation. The viral protein Rev controls the nucleocytoplasmic export of unspliced and singly-spliced RNA that is central to proviral replication-competence and is therefore a prerequisite for efficient viral reactivation during the "shock-and-kill" cure therapy. Here we show that during infection and reactivation, unspliced HIV-1 RNA is a subject to complex and dynamic regulation by the Rev cofactor MATR3 and the MTR4 cofactor of the nuclear exosome. MATR3 and MTR4 coexist in the same ribonucleoprotein complex functioning to either maintain or degrade the RNA, respectively, with Rev orchestrating this regulatory switch. Moreover, we provide evidence of nuclear retention of unspliced HIV-1 RNA in ex vivo cultures from 22 ART-treated people with HIV, highlighting a reversible post-transcriptional block to viral RNA nucleocytoplasmic export that is relevant to the design of curative interventions.

HIV-1 replication can be effectively blocked in people with HIV (PWH) by antiretroviral therapy (ART), turning the life-threatening disease into a chronic illness[1]. However, ART is not curative and has to be taken for life. Consequently, HIV-1 cure must be achieved: either remission (durable control of the virus in the absence of ART) or eradication (complete removal of intact and replication-competent virus). The persistence of latently infected cells—a minute fraction of host cells that harbors integrated, intact proviruses that are replication-competent, and thus form the HIV-1 reservoir, is a major obstacle to reaching a cure[2,3]. Although many cell types may contribute to the latent

[1]Laboratory of Molecular Virology, Malopolska Centre of Biotechnology, Jagiellonian University, Krakow, Poland. [2]Doctoral School of Exact and Natural Sciences, Jagiellonian University, Lojasiewicza 11, 30-348, Krakow, Poland. [3]Service of Molecular Virology, Department of Molecular Biology (DBM), Université Libre de Bruxelles (ULB), Gosselies, Belgium. [4]Laboratory of Experimental Virology, Department of Medical Microbiology, Amsterdam UMC, University of Amsterdam, Amsterdam, Netherlands. [5]Dept of Molecular Medicine and Medical Biotechnology, Università degli Studi di Napoli "Federico II", Naples, Italy. [6]Department of Infectious Diseases and Tropical Medicine, Jagiellonian University Medical College, Krakow, Poland. [7]Université Paris Cité, INSERM U1016, CNRS UMR8104, Institut Cochin, Paris, France. [8]CHU d'Orléans, Orléans, France. [9]Université d'Orléans, LI²RSO, Orléans, France. [10]Service des Maladies Infectieuses, CHU St-Pierre, Université Libre de Bruxelles (ULB), Brussels 1000, Belgium. [11]Laboratory of Molecular Virology, The International Centre for Genetic Engineering and Biotechnology (ICGEB), Trieste, Italy. [12]Virogenetics Laboratory of Virology, Malopolska Centre of Biotechnology, Jagiellonian University, Krakow, Poland. [13]These authors contributed equally: Agnieszka Dorman, Maryam Bendoumou. [14]These authors jointly supervised this work: Alexander O. Pasternak, Carine Van Lint, Anna Kula-Pacurar. ✉e-mail: k.a.pyrc@uj.edu.pl; a.o.pasternak@amsterdamumc.nl; carine.vanlint@ulb.be; anna.kula-pacurar@uj.edu.pl

reservoir, including monocytes, monocytic-derived macrophages, or microglial cells[4,5], the best characterized is a small population of long-lived HIV-infected resting memory CD4+ T cells that clonally proliferate due to cellular responses to antigens, homeostatic drivers, or specific HIV-1 integration sites[6,7]. Latently infected cells are largely insensitive to ART and to the recognition by the immune system. However, the state of viral latency is reversible and latently infected cells can be reactivated to produce virus following cellular activation, thereby providing a potential source of viremia once ART is interrupted.

Several therapeutic strategies to eliminate HIV-1 latency have emerged, and one of them, referred to as "shock and kill", has been extensively studied[8]. This strategy aims at reactivating latently infected cells by using small chemical compounds called latency-reversing agents (LRAs) (the "shock" phase) while boosting the immune responses to eliminate the reactivated infected cells (the "kill" phase). Over the years, increasing knowledge of HIV-1 latency has led to the emergence of several classes of LRAs that reactivate HIV-1 transcription[9,10]. These include epigenetic LRAs inhibiting histone deacetylases (HDACIs) such as SAHA, panobinostat, and romidepsin[11–13]; histone methyltransferases inhibitors (chaetocin and BIX0194)[14]; DNA methyltransferase inhibitors or demethylating agents (5-AzadC)[15]. LRAs also include protein kinase C (PKC) agonists inducing NF-κB such as prostratin, bryostatin, and ingenols[14,15]; bromodomain and extra terminal domain inhibitors BETi inducing P-TEFb such as JQ1, I-BET, and I-BET151[16], activators of the Akt pathway such as disulfiram[17], STAT5 sumoylation inhibitors[18], SMAC mimetics[19] and different immunomodulatory LRAs (reviewed in ref. 20). Single or combined LRA treatments have been shown to reverse latency in in vitro and ex vivo latency models[15,16,21,22]. In vivo, however, LRAs have been unable to reduce the number of latently infected cells[11,12] or delay the time to viral rebound following cessation of ART[23]. Notably, most of the in vivo reactivation studies have demonstrated a rather weak reactivation effect of LRAs, i.e., increases were only observed in cell-associated viral RNA expression with a much lower or no effect on HIV-1 RNA levels in plasma, as shown for SAHA and disulfiram[11,24,25]. These studies suggest that post-transcriptional blocks may mitigate the effect of certain LRAs as they act by relieving transcriptional blocks and have no known impact on post-transcriptional blocks.

Traditionally, latent reservoir was defined as transcriptionally silent due to epigenetic and transcription repression mechanisms including (i) DNA methylation and diverse post-translational modifications of histones; (ii) the absence of host transcriptional activators and of viral Tat; (iii) presence of transcriptional repressors such as CTIP2[26]; (iv) sequestration into 7SK of P-TEFb and (v) transcriptional interference (reviewed in ref. 27). Also, proviral integration into transcriptionally silent chromatin context or subnuclear spatial localization of HIV-1 integration sites that is repressive for transcription were reported[28,29]. It is now apparent, due to the development of more sensitive techniques, that cell-associated viral RNAs or even proteins are detected in ART-treated PWH without ex vivo stimulation[30–32]. Moreover, the elegant works by the Yukl group have challenged the dogma that HIV-1 latency is mainly regulated at the level of transcriptional initiation, as it has revealed several additional blocks to transcriptional elongation, polyadenylation, and splicing[33–35], suggesting that latency can be also maintained by less-characterized mechanisms operating at the post-transcriptional level. These may include (i) splicing defects[36]; (ii) nuclear retention of multiply-spliced (MS) HIV-1 RNA in resting CD4 + T cells[37]; (iii) inhibition of expression of viral proteins in a codon-usage dependent manner[38], (iv) inhibition of HIV-1 translation by microRNAs[39] and (v) multiple biochemical and metabolic blocks that are not completely released by some of the LRAs[40]. We have recently measured low expression levels of nuclear protein MATR3, a known co-factor of the viral protein Rev[41,42] in quiescent PWH cells, highlighting the relevance of a post-transcriptional latency block

linked to RNA export[43]. HIV-1 expresses three classes of RNA: intron-containing US and singly-spliced (SS) RNA, as well as intronless MS transcripts. MS HIV-1 RNA encodes the regulatory viral proteins Tat and Rev with the latter promoting the nucleocytoplasmic export of SS and US HIV-1 transcripts. SS and US HIV-1 RNA represent a template for structural gene production, and US HIV-1 RNA is also the full-length genomic RNA that is encapsidated into newly formed virions[44]. Rev is a critical viral factor for productive viral infection and, thus, a critical component of replication-competent proviruses that are responsible for viral rebound. Therefore, Rev is required for purging the virus in the "shock-and-kill" strategy.

In this study, we shed light on the regulatory mechanisms that control the fate of the Rev-dependent unspliced HIV-1 RNA and define a previously uncharacterized post-transcriptional block in Rev-dependent export that is relevant to viral latency and reactivation.

## Results

### Rev co-factor MATR3 stabilizes HIV-1 RNA during reactivation

Little is known about the post-transcriptional fate of unspliced genomic HIV-1 RNA in latency and reactivation. We have previously shown that limiting levels of Rev co-factor MATR3[41] correlate with viral reactivation from latency, highlighting the relevance of post-transcriptional mechanisms contributing to latency and reactivation[43]. To gain further insights into the mechanism of MATR3 regulation, we depleted this factor in J-Lat 9.2 cellular model of latency and measured the Rev-dependent and Rev-independent HIV-1 expression. J-Lat 9.2 cells harbor a near-full-length latent HIV-1 provirus, in which *gfp* reporter gene (inserted in place of *nef*) is expressed from MS transcripts. Nucleocytoplasmic export of MS RNA and consequently, GFP expression in J-Lat 9.2 cells, are independent of Rev activity. On the other hand, extracellular HIV-1 RNA represents virus production that depends on Rev activity. Here, we depleted MATR3 using previously published shRNA technology[43] from untreated or TNFα-treated J-Lat 9.2 cells. As shown in Fig. 1a, MATR3 knock-down did not affect the percentages of GFP-positive cells (Supplementary Fig. 1). However, it decreased the extracellular genomic HIV-1 RNA levels (Fig. 1b), which is in line with an inhibition of the Rev function. Similarly, MATR3 depletion caused decreases in the total levels of Rev-dependent RRE-containing transcripts but did not change the level of Rev-independent MS HIV-1 RNA (Supplementary Fig. 2a, b) highlighting the role of MATR3 in viral reactivation. Therefore, to pinpoint the step during reactivation at which MATR3 acts on Rev-dependent viral transcripts, we assessed the nuclear and cytoplasmic levels of viral RNAs. MATR3 knock-down and purity of cytoplasmic and nuclear fractions were confirmed by Western blot (Fig. 1c). Next, analysis of viral transcripts in both fractions revealed a reduction in both nuclear and cytoplasmic TNFα-induced RRE HIV-1 RNA (Fig. 1d) but no change in the nuclear and cytoplasmic MS HIV-1 RNA levels when compared to control shLuc-transduced cells (Fig. 1e). This phenotype was further confirmed by Immuno-RNA FISH protocol for Rev-dependent HIV-1 ᵍᵃᵍRNA and lamin B1 (to visualize nuclear membrane) to detect viral transcripts inside and outside the nucleus. As shown in Supplementary Fig. 2c, RNA FISH revealed several spots of HIV ᵍᵃᵍRNA with diverse sizes inside and outside of the nucleus. Further analysis of confocal images revealed a reduction in the number (Fig. 1f) and the volume (Fig. 1g) of nuclear and cytoplasmic viral ᵍᵃᵍRNA foci upon MATR3 depletion, suggesting its nuclear role in stabilization of Rev-dependent transcripts, which we next confirmed with the actinomycin D treatment of TNFα-stimulated J-Lat 9.2 cells (Fig. 1h, i). Of note, when the function of Rev was inhibited with ABX464 compound[45] in TNFα-treated J-Lat 9.2 cells, viral reactivation was impaired, similarly to the MATR3 depletion phenotype (Supplementary Fig. 3a–c). However, an enrichment in nuclear levels of RRE-containing transcripts was observed as assessed by RT-qPCR and immuno-RNA FISH (Supplementary Fig. 3d–g), which is in line with nucleocytoplasmic export block.

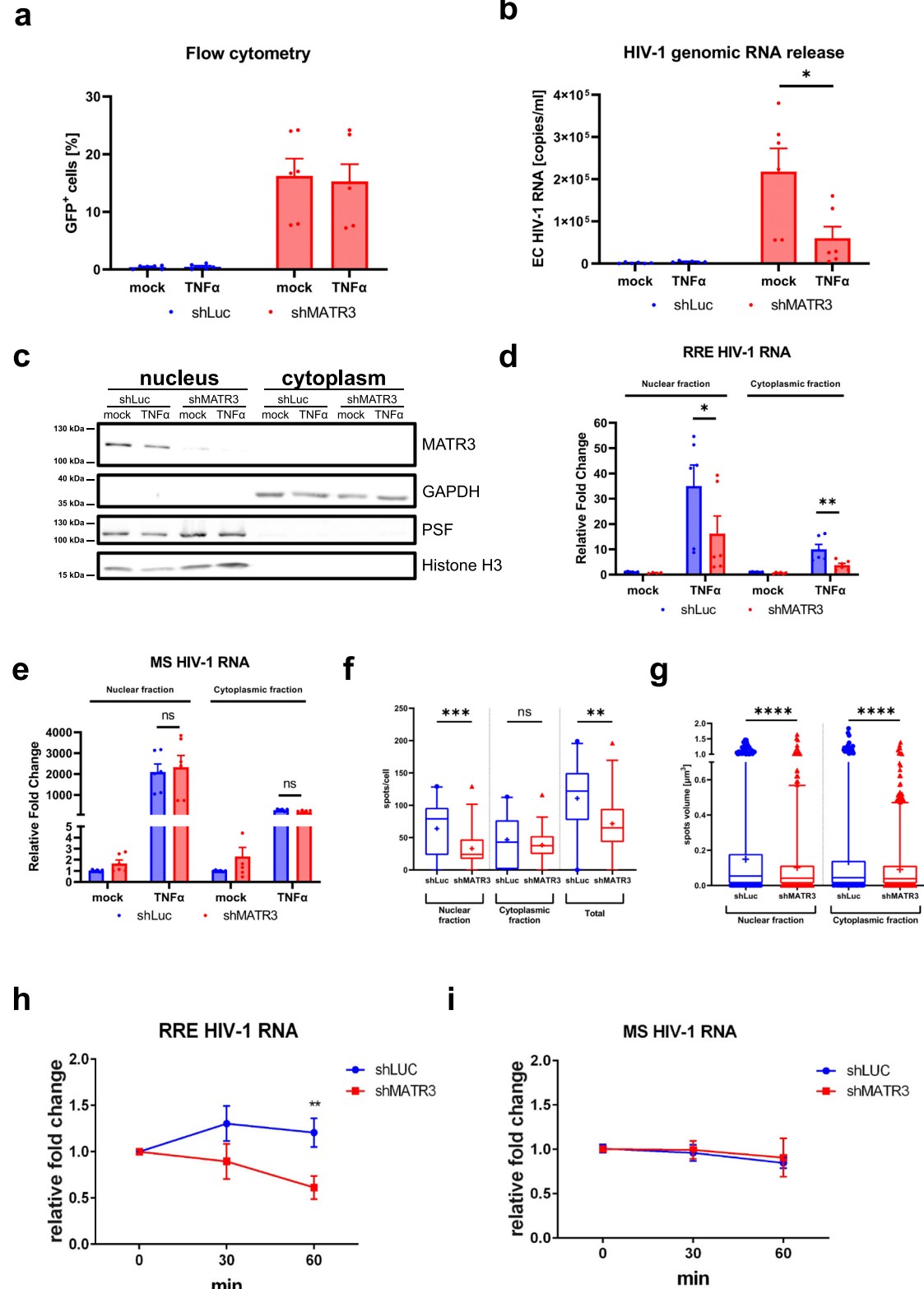

In conclusion, our results are consistent with a scenario in which, during reactivation, Rev-dependent transcripts are stabilized by MATR3 in the nucleus, a step that is required for Rev-mediated nuclear export.

### MATR3/MTR4/Rev ribonucleoprotein complex regulates the fate of Rev-dependent HIV-1 RNA

Results presented above suggest that MATR3 may protect HIV-1 RNA from the RNA degradation machinery in the nucleus. MTR4 is an essential exosome co-factor that directs aberrant or intron-containing RNAs to exosome complex for degradation[46] and a previous study has identified MATR3 in the MTR4 proteome screen[47]. First, we wished to confirm the interaction of these proteins with HIV-1 RNA. We used our previously published MS2-tagged viral vector that allows affinity purification of HIV-1 RNA with associated factors by flag-MS2 pull down[41]. Hence, 293 T cells were transfected with pHIV-INTRO viral vector and plasmids expressing flag-MS2 and the viral protein Tat to activate viral

**Fig. 1 | MATR3 stabilizes Rev-dependent HIV-1 RNA during reactivation.** MATR3 depletion in J-Lat 9.2 cells was obtained with shMATR3 and compared to shLuc lentiviral transduction and three days after puromycin selection, cells were treated with TNFα [10 ng/ml] and after 24 h subjected to **a** flow cytometry to quantify the percentages of GFP⁺ cells, **b** RNA isolation from the supernatant followed by RT-qPCR for genomic HIV-1 RNA (primers for *gag RNA*), **c**–**e** biochemical fractionation protocol to obtain nuclear and cytoplasmic extracts, **f**–**g** immuno-RNA FISH followed by confocal microscopy analysis or **h**, **i** RNA stability test with actinomycin D (ActD). **c** Immunoblotting using anti-MART3, anti-histone H3, and anti-PSF as chromatin and nucleoplasm markers, respectively, and anti-GAPDH as a cytoplasmic marker. **d**, **e** Samples were subjected to RNA isolation followed by RT-qPCR for RRE-containing (**d**) and MS (**e**) HIV-1 transcripts. Values were normalized using 18S RNA primers and were presented as relative fold changes to the values measured in the untreated shLuc nuclear and cytoplasmic samples which were arbitrarily set at a value of 1. The number (**f**) and volume (**g**) of nuclear, cytoplasmic, and total ᵍᵃᵍHIV-1 spots were quantified from z-stacks obtained from 10 images/biological repetition, $n = 3$. Results are presented as box and whiskers with 5–95% confidence interval. Median value is shown as a bar, dots are points outside whiskers representing outliers, and the mean value is shown as "+". Statistics were performed using a two-tailed unpaired Student's $t$ test. Statistical comparisons are indicated if $p \leq 0.01$ (**), $p \leq 0.001$ (***), and $p \leq 0.0001$ (****), "ns" indicate no significant. **h**–**i** Cells were collected at 0, 30, and 60 min and subjected to RNA isolation following RT-qPCR targeting RRE-containing (**h**) and MS (**i**) HIV-1 RNA. Values were normalized using 18S RNA primers and further normalized to DMSO control at each time point. All results from **a**, **b**, **d**, **e**, **h**, **i** are shown as mean values ± SEM, $n = 3$ (**a**, **b**, **d**, **e**) and $n = 4$ (**h**, **i**) biological replicates in duplicates. Statistics were performed using a two-tailed paired Student's $t$ test (**a**, **b**, **d**, **e**, **h**, **i**). Statistical comparisons are indicated if $p \leq 0.5$ (*), $p \leq 0.01$ (**), $p \leq 0.001$ (***), and "ns" indicate not significant. Source data are provided as a Source Data file.

transcription. As shown in Fig. 2a, Tat, MATR3, and PSF that are known HIV-1 RNA interactors[42] as well as MTR4, an exosome RNA helicase and EXOSC-10, an exosome catalytic subunit, were specifically bound to HIV-1 RNA (Fig. 2a). As MATR3 interacts with Rev[41,48], we next assessed the possible interaction between MTR4 and MATR3/Rev by immuno-precipitating MATR3 from 293 T cells that were co-transfected with flag-MATR3, viral vector, and plasmids expressing Tat and Rev-GFP in the presence and absence of nuclease treatment. As shown in Fig. 2b, MATR3 bound MTR4 via protein-protein interactions while interactions between MATR3 and Rev were mediated by RNA as also shown previously[41,48]. Similarly, immunoprecipitation of endogenous MTR4 from 293 T cells, co-transfected and treated with or without nuclease as described above, demonstrated a protein-protein interaction with MATR3, while the interaction with Rev was RNA-dependent (Fig. 2c). These results suggest that the three proteins form a ribonucleoprotein complex. To test this hypothesis, we conducted a re-IP experiment using 293 T cells co-transfected with flag-MATR3, viral vector, and plasmids expressing Tat and Rev-GFP. IP against flag-MATR3 was followed by elution with flag peptides for subsequent re-IP against Rev-GFP. As shown in Fig. 2d, MTR4 was present in re-IP against Rev-GFP, demonstrating the existence of MATR3/MTR4/Rev complex. We also addressed the colocalization between MATR3 and MTR4 with respect to HIV-1 RNA by immuno-RNA FISH in TNFα-activated J-Lat 9.2 cells. As shown in Fig. 2e, MATR3 and MTR4 colocalized with each other in several spots in the nucleus. When we stained viral RNA, we could appreciate multiple RNA spots including one large bright spot corresponding to viral transcription site as described previously[28,29,42] and we showed that MATR3 and MTR4 colocalized with HIV-1 ᵍᵃᵍRNA in few spots in the nucleoplasm (Fig. 2e). Quantification of HIV-1 RNA-MATR3, HIV-1 RNA-MTR4, and MATR3-MTR4 colocalizations revealed a 30%, 65%, and 18% overlap of spots, respectively, while their volume overlap was 41%, 43%, and 34%, respectively (Fig. 2f). Next, HIV-1 RNA-MATR3-MTR4 triple colocalization quantification showed 19% overlap between the spots and 39% overlap between their volume (Fig. 2f) supporting the idea of a transient interaction between MATR3 and MTR4 with HIV-1 RNA.

Next, to explore the potential role of MTR4 in controlling the fate of Rev-dependent transcripts, we addressed its impact on viral RNAs in the presence and absence of Rev. Depletion of MTR4 caused increases in the levels of all viral transcripts measured (TAR, RRE and MS HIV-1 RNA) highlighting the known role of MTR4 in HIV-1 transcriptional repression[47] (Fig. 2g, Supplementary Fig. 4a). Moreover, in the presence of Rev, we observed an additional increase in the levels of RRE-containing HIV-1 transcripts, which suggested a post-transcriptional negative role of MTR4 in Rev-dependent pathway (Fig. 2g, compare lane 7 with lane 8). Furthermore, fractionating the cells revealed that in the absence of Rev, depletion of MTR4 caused increases in the nuclear levels of RRE-containing transcripts. However, in the presence of Rev, a strong increase in their cytoplasmic levels was observed, suggesting an augmented Rev activity when MTR4 was missing (Fig. 2h). Next, to functionally link MTR4 with MATR3, we performed RNA IP (RIP) against endogenous MATR3 on MTR4-depleted 293 T cells co-transfected with viral vector, Tat, and Rev. MTR4 was depleted and endogenous MATR3 was immunoprecipitated successfully (Supplementary Fig. 4b, c). Depletion of MTR4 caused increased RRE-containing HIV-1 RNA binding to MATR3 (Fig. 2i) but did not change the *gapdh* RNA binding (Supplementary Fig. 5d), highlighting the opposing activities of these two factors in the regulation of HIV-1 RNA.

Together, these data underscore that MATR3/MTR4/Rev ribonucleoprotein complex regulates the fate of Rev-dependent HIV-1 RNA.

## Rev determines the MATR3-MTR4 binding to HIV-1 RNA

Next, we went on to explore the Rev dependence by determining how MATR3 and MTR4 interact with HIV-1 RNA in a more physiological model of HIV-1 infection. To this end, we used a primary CD4⁺ T-cell model of infection with a full-length HIV-1 molecular clone pNL4.3 mutated in the Rev open reading frame or with wild-type (wt) pNL4.3[49]. Infection with pNL4.3ΔRev did not change the integration frequency as compared to pNL3.4 (Fig. 3a). As expected, Rev-dependent transcripts significantly decreased in pNL4.3ΔRev-infected primary cells, but we also observed a smaller reduction in *tat*-containing (Rev-independent) HIV-1 RNA (Fig. 3b). However, export of Rev-independent transcripts was not affected (Fig. 3c). Importantly, cytoplasmic relative levels of Rev-dependent transcripts decreased in pNL4.3ΔRev versus wt pNL4.3 (Fig. 3d), underscoring impairments in nucleocytoplasmic export. Next, to address the dependence of Rev in MATR3-MTR4 opposing interplay on viral RNA, we performed RIP protocol using anti-MATR3 and anti-MTR4 antibodies from either pNL4.3ΔRev- or wt pNL4.3-infected primary CD4⁺ T cells. As shown in Fig. 3e, the absence of Rev significantly decreased the MATR3 interaction with Rev-dependent transcripts. However, interaction with MTR4 did not change in pNL4.3ΔRev-infected cells (Fig. 3e) underscoring the complexity of molecular mechanisms in primary cells. Thus, we infected Jurkat CD4⁺ T cells with either pNL4.3 or pNL4.3ΔRev. Levels of integrated HIV-1 DNA remained unchanged between the two infections (Fig. 3f). As expected, Rev-dependent transcripts significantly decreased in pNL4.3ΔRev-infected Jurkat cells (Fig. 3g), but we also observed an increase in *tat*-containing (Rev-independent) HIV-1 RNA (Fig. 3g), which was in line with the reported role of Rev in splicing inhibition[50]. Subsequent RIP against MTR4 showed significant increases in MTR4 binding to Rev-dependent transcript when Rev was missing (Fig. 3h). The pronounced MTR4 phenotype in Jurkat cells over primary CD4⁺ T cells highlights the complex and heterogenous nature of molecular mechanisms that contribute to the HIV-1 gene expression in primary cells. Altogether, the results above show that Rev determines the MATR3-MTR4 interplay in Rev-dependent HIV-1 RNA regulation.

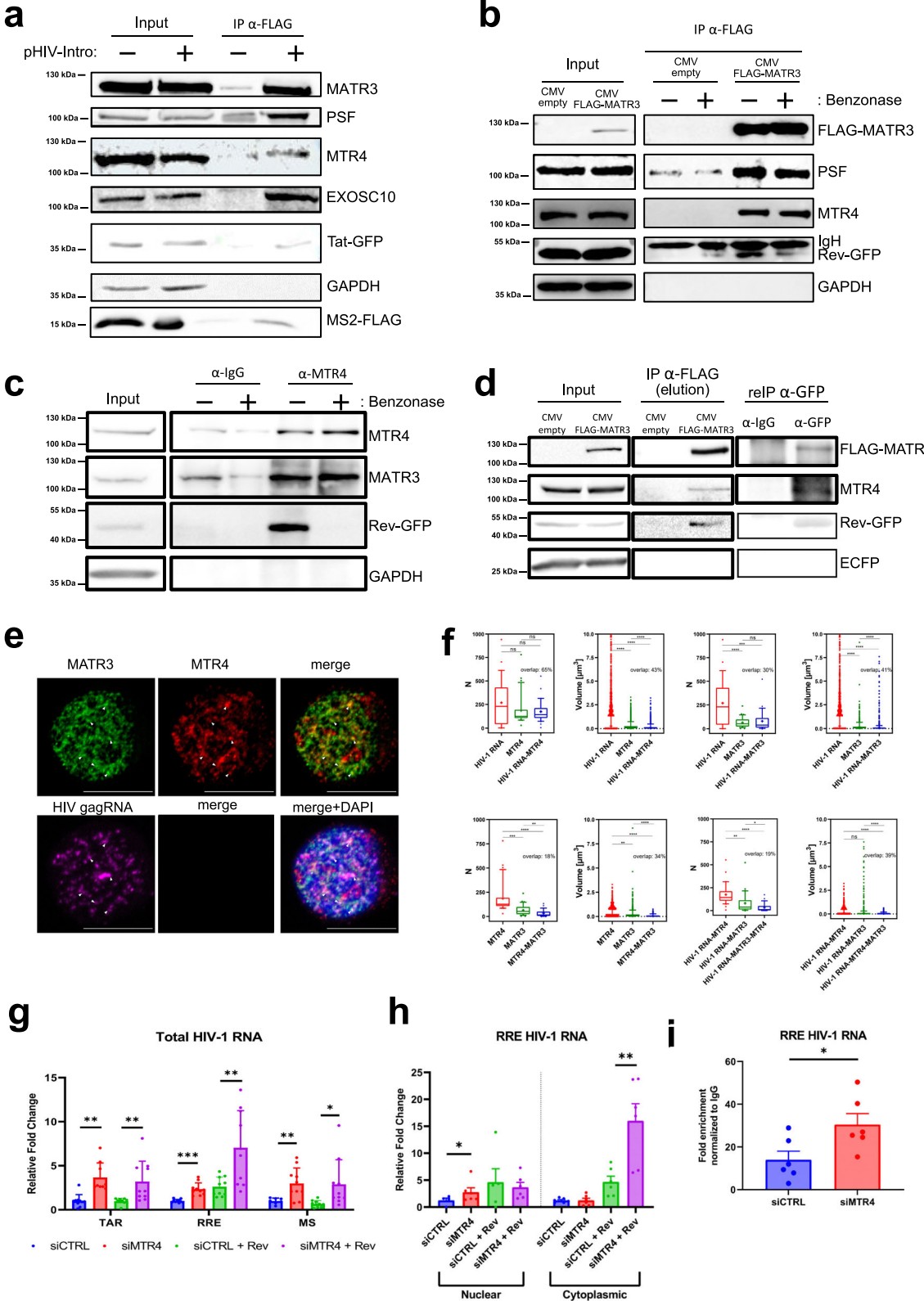

## Nuclear retention of US HIV-1 RNA in PBMCs from ART-treated PWH

The opposing nature of the interaction between MATR3 and MTR4 with viral RNA triggered by Rev either stabilizing or degrading viral transcripts may shed light on post-transcriptional mechanisms that could be crucial for maintaining HIV-1 latency or enabling viral reactivation. The varying levels of these two factors in unstimulated versus

activated cells may shift the balance, favoring one of these scenarios. Previously, we have reported limiting levels of MATR3 in CD8+-depleted PBMCs from ART-treated PWH that correlated with the limited ability of some LRAs to reactivate the virus[43]. We, therefore, evaluated the levels of MATR3 and MTR4 in ex vivo cultures of PBMCs isolated from 3 healthy donors that were either unstimulated or activated with PHA. As previously published, MATR3 levels were limited in

**Fig. 2 | MATR3/MTR4/Rev ribonucleoprotein complex regulates the fate of Rev-dependent HIV-1 RNA. a** 293 T cells were transfected with MS2-tagged pHIV-INTRO, pTat-GFP, and pMS2-FLAG and after 24 h whole cell lysates were subjected to anti-FLAG immunoprecipitation. **b**, **c** 293 T cells were transfected with pFLAG-MATR3, pHIV-INTRO, pTat, and pRev-GFP plasmids and after 24 h whole cell lysates were subjected to anti-FLAG (**b**), anti-MTR4/anti-IgG (**c**) immunoprecipitation followed by ±benzonase treatment or (**d**) anti-FLAG immunoprecipitation followed by elution with FLAG peptides for subsequent IP using either anti-IgG or anti-GFP antibodies. GAPDH is a loading control in input samples and indicates the specificity of IP. ECFP expressed from pHIV-INTRO is a transfection efficiency control in input samples and indicates the specificity of IPs. IgH (in **b**) refers to the immunoglobulin heavy chain from anti-flag mouse antibodies that are attached to beads. This was detected with a secondary anti-mouse antibody during the anti-GFP immunoblotting. **e** J-Lat 9.2 cells were stimulated with TNFα [10 ng/ml] and subjected to RNA FISH and immunostaining using antibodies against MTR4 and MATR3 for subsequent confocal microscopy analysis. MATR3 is shown in green, MTR4 in red, [gag]HIV-1 RNA in purple, and DAPI-stained nucleus in blue. Bright, large [gag]HIV-1 RNA spot corresponds to viral transcription site as described previously[28,29,42]. White arrows indicate triple colocalization sites. Scale bar = 10 μm. **f** Colocalization between MATR3-HIV-1 RNA, MTR4-HIV-1 RNA, MATR3-MTR4, and

MATR3-MTR4-HIV-1 RNA was quantified by counting the number of colocalizing spots and their volumes from z-stacks obtained from 10 images/biological repetition, $n = 3$. Results are presented as box and whiskers with 5–95% confidence interval. Median value is shown as a bar, dots are points outside whiskers representing outliers, mean value is shown as "+". Statistics were performed using a two-tailed unpaired Student's $t$ test. Statistical comparisons are indicated if $p \leq 0.01$ (**), $p \leq 0.001$ (***), and $p \leq 0.0001$ (****), "ns" indicate no significance. **g–i** MTR4 depletion in 293 T cells was obtained using siRNA transfection. After 24 h, cells were transfected with plasmids pHIV-INTRO, pTat-GFP, and pRev-GFP and after 24 h collected for (**g**) total RNA isolation, (**h**) nuclear-cytoplasmic fractionation protocol followed by RT-qPCR for RRE-containing, TAR, and MS HIV-1 RNAs normalized using *gapdh* primers and presented as relative fold changes to the values measured in non-targeting control (siCTRL) condition, which was arbitrarily set at a value of 1. **i** total RNA isolation followed by RNA immunoprecipitation (IP) protocol using anti-MATR3 and anti-IgG isotype antibodies. RNA was purified from IPs and subjected to RT-qPCR targeting RRE-containing HIV-1 RNA. Values were normalized to input and shown as fold enrichment over IgG control. All results are shown as mean values ± SEM from $n = 5$ (**g**), $n = 3$ (**h**), $n = 3$ (**i**) biological replicates in duplicates. Statistics were performed using a two-tailed paired Student's $t$ test, $p \leq 0.5$ (*), $p \leq 0.01$ (**), $p \leq 0.001$ (***). Source data are provided as a Source Data file.

unstimulated cells and upregulated upon PHA treatment (Fig. 4a)[43]. Notably, MTR4 levels were abundant in both conditions (Fig. 4a). These varying levels of both factors in unstimulated cells suggest that the export of the viral RNA might be impaired. To test this, freshly isolated PBMC from the blood of 16 ART-treated PWH with undetectable viral loads in plasma (Supplementary Table 1) were subjected to nucleocytoplasmic fractionation protocol. Total nucleic acids were separately extracted from the nuclear and cytoplasmic fractions, and the purity of the cytoplasmic fraction was assessed by measuring the levels of β-actin DNA. Of the total cellular β-actin DNA, only a median of 0.05% was detectable in the cytoplasm, indicating the high purity of the cytoplasmic fraction (Supplementary Fig. 5a). Next, we measured the levels of US HIV-1 RNA in the nuclear and cytoplasmic fractions. Because of possible latency blocks to HIV-1 transcription elongation resulting in gradient of unspliced HIV-1 transcript abundance[33], we used two sets of primers detecting shorter (amplicon spanning the packaging signal and the beginning of *gag* ORF, minimal transcript length 342 nt) and longer (amplicon located in the middle of *gag* ORF, minimal transcript length 1045 nt) US HIV-1 transcripts (hereafter termed US-short and US-long, respectively). In parallel, we measured levels of cellular *gapdh* mRNA. Levels of US-short RNA were comparable between the nuclear and cytoplasmic fractions, and the same subcellular distribution was observed for *gapdh* mRNA (Fig. 4b, c). On the other hand, levels of US-long RNA were significantly higher in the nucleus than in the cytoplasm ($p = 0.0078$), indicating that in uncultured ex vivo PBMCs, most of the long US transcripts were retained in the nucleus (Fig. 4d). Indeed, medians of 46.0% of US-short RNA and 49.4% *gapdh* mRNA were present in the cytoplasmic fraction, as opposed to US-long RNA, of which only a median of 4.8% was present in the cytoplasm (Fig. 4e). We also measured MS HIV-1 RNA in both fractions but could detect this transcript in the nuclear fractions from one participant only and in none of the cytoplasmic fractions (Supplementary Fig. 5b), in agreement with previous studies showing low levels of MS HIV-1 RNA in PWH on ART[51–53].

We next evaluated the observed post-transcriptional block in untreated and LRA-treated ex vivo cultures. To this end, ex vivo cultures of CD8[+]-depleted PBMCs isolated from the peripheral blood of 6 ART-treated PWH with undetectable viral loads in plasma (Supplementary Table 1) were mock-treated or treated with SAHA [0.5 μM], disulfiram [0.5 μM], romidepsin [17.5 nM] or PHA [1.66 μg/ml] in the presence of antiretrovirals to prevent viral spread in the cell cultures. After 3 days post-treatment, cells were subjected to nucleocytoplasmic fractionation. The purity of fractions was assessed by Western blot and by measurements of the levels of β-actin DNA (Supplementary

Fig. 5c, d). As above, only a minute fraction of total β-actin DNA was present in the cytoplasm, indicating the high purity of the cytoplasmic fraction (Supplementary Fig. 5d). Next, we measured the levels of nuclear and cytoplasmic HIV-1 US-short and US-long transcripts, along with cellular *gapdh* mRNA. Levels of US-short and *gapdh* mRNA were largely comparable between the fractions either in the unstimulated or LRA-treated conditions (Fig. 4f–i). On the other hand, only a median of 1.8% of US-long RNA in the mock condition was present in the cytoplasm, highlighting that US-long RNA was retained in the nucleus in unstimulated ex vivo cultures (Fig. 4j, k). Notably, LRA treatment did not overcome the nuclear retention of US-long RNA, except for romidepsin which was able to reverse the block as a median of 71.0% of US-long RNA was present in the cytoplasmic fraction upon romidepsin treatment (Fig. 4k). As above, we could detect MS HIV-1 RNA only in the minority of cultures, but in this limited number of participants with detectable MS RNA, we did not observe any nuclear retention of this transcript (Supplementary Fig. 5e, f).

Summarizing all findings presented here above, we propose a working model of highly dynamic regulatory mechanisms that determine the post-transcriptional fate of US HIV-1 RNA (Fig. 4l). Limiting levels of MATR3 and Rev in unstimulated conditions may cause nuclear retention of unspliced HIV-1 RNA that is destabilized by MTR4, whose level is abundant. During HIV-1 reactivation, MATR3 and Rev hijack the RNA to stabilize and export the RNA, respectively. This mechanism is likely crucial for maintaining HIV-1 latency or enabling viral reactivation.

## Discussion

This study sheds light on the regulatory mechanisms that control the fate of the Rev-dependent unspliced HIV-1 RNA. We also report a previously uncharacterized post-transcriptional block in nucleocytoplasmic export, relevant to HIV-1 latency and reactivation, that causes nuclear retention of US HIV-1 RNA.

Our results indicate that MATR3/MTR4/Rev ribonucleoprotein complex regulates the fate of Rev-dependent viral transcripts. MTR4 RNA helicase plays an essential role in all aspects of nuclear exosome function by recognizing exosome target RNAs, recruiting them to the exosome, and facilitating their channeling into the exosome[46,54]. Intron-containing (or aberrant) RNAs are subject to nuclear quality control mechanisms that discriminate between correctly processed and aberrant RNAs in order to ensure the integrity of gene expression[55]. US and SS HIV-1 transcripts contain introns, and thus, they might be subject to RNA degradation[56]. Recent studies have reported that MTR4 and other components of the nuclear surveillance

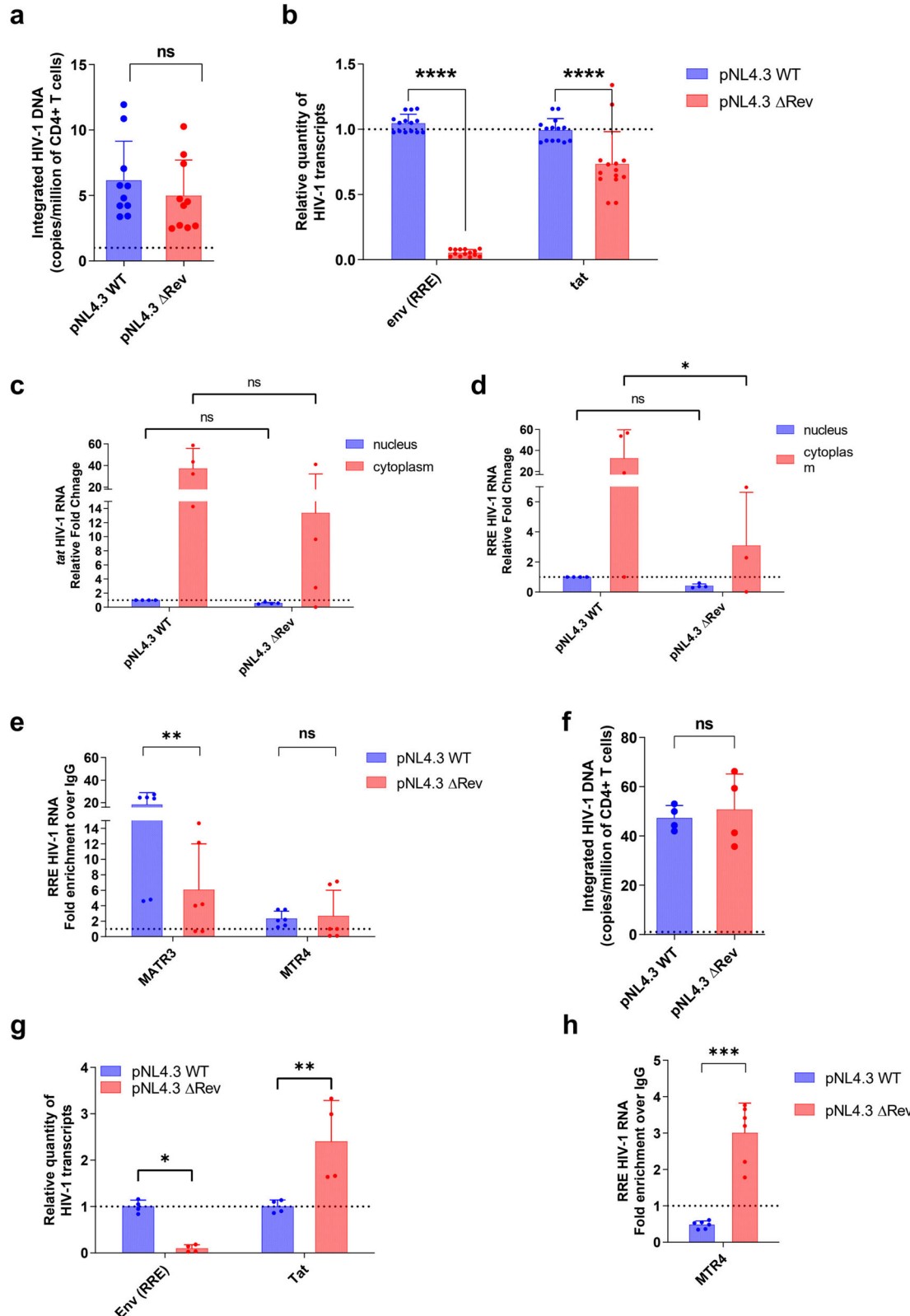

complex suppress HIV-1 transcription initiation[47,57]. We showed that the knock-down of MTR4 in the presence of Rev increased total and cytoplasmic levels of RRE-containing transcripts and caused an increase in the interaction of this RNA with MATR3, as shown by RIP. Thus, MTR4 seems to also play a post-transcriptional repressive role during HIV-1 RNA processing.

Our subsequent RIP experiments in infections with full-length pNL4.3 mutated in Rev support the opposing activities of the two

factors on the viral RNA that are Rev-dependent. Opposing activities of MATR3 and MTR4 in a complex suggest that they may act as "molecular switches". Interestingly, the recent study of Wang et al. has demonstrated a protein-protein interaction between MTR4 and NRDE2 with opposing activities to control the stability of mRNA[58]. NRDE2 was shown to lock MTR4 in an inactive state to assure mRNA stability and export[58]. Our data suggest that MATR3-MTR4 complex may act as "molecular switch" triggered by Rev to ensure rapid and

**Fig. 3 | Rev determines the MATR3 and MTR4 binding to viral RNA.** Primary CD4+ T cells were isolated from blood of healthy donors and infected with WT pNL4.3 HIV particles or Rev-mutated pNL4.3 HIV particles and collected after 3 days and used for (**a**) DNA extraction to quantify total HIV DNA by qPCR, for (**b**) total RNA extraction used for RT-qPCR quantification of RRE-containing RNA and Tat HIV-1 RNA, for (**c**) nuclear-cytoplasmic fractionation protocol followed by quantitative real-time RT-PCR for Tat and (**d**) RRE HIV-1 RNA, for (**e**) RNA immunoprecipitation using anti-MATR3, anti-MTR4 and anti-IgG isotype antibodies. RNA was purified from IPs and subjected to RT-qPCR targeting RRE-containing HIV-1 RNA. Values were normalized to input and shown as fold enrichment of IgG control. **f**–**h** Jurkat T cells were infected with WT pNL4.3 HIV particles or Rev-mutated

pNL4.3 HIV particles and collected after 3 days and used for (**f**) DNA extraction to quantify total HIV DNA by qPCR, for (**g**) total RNA extraction used for RT-qPCR quantification of RRE-containing and Tat HIV-1 RNA and (**h**) RNA immunoprecipitation using anti-MTR4 and anti-IgG isotype antibodies. RNA was purified from IPs and subjected to RT-qPCR targeting RRE-containing HIV-1 RNA. All results are shown as mean values ± SEM from $n = 5$ (**a**), $n = 7$ (**b**), $n = 4$ (**c**, **d**, **f**, **g**), $n = 6$ (**e**), $n = 3$ (**h**) biological replicates in duplicates. Statistics were performed using a two-sided Wilcoxon signed-rank test for HIV integration (**a**), a two-way ANOVA test (**b**–**g**), and a two-tailed unpaired Student's $t$ test (**h**). Significant $p$ values are indicated by the asterisks above the graphs $p \le 0.05$ [*], $p \le 0.01$ [**], $\le 0.001$ [***], $\le 0.0001$ [****]. Source data are provided as a Source Data file.

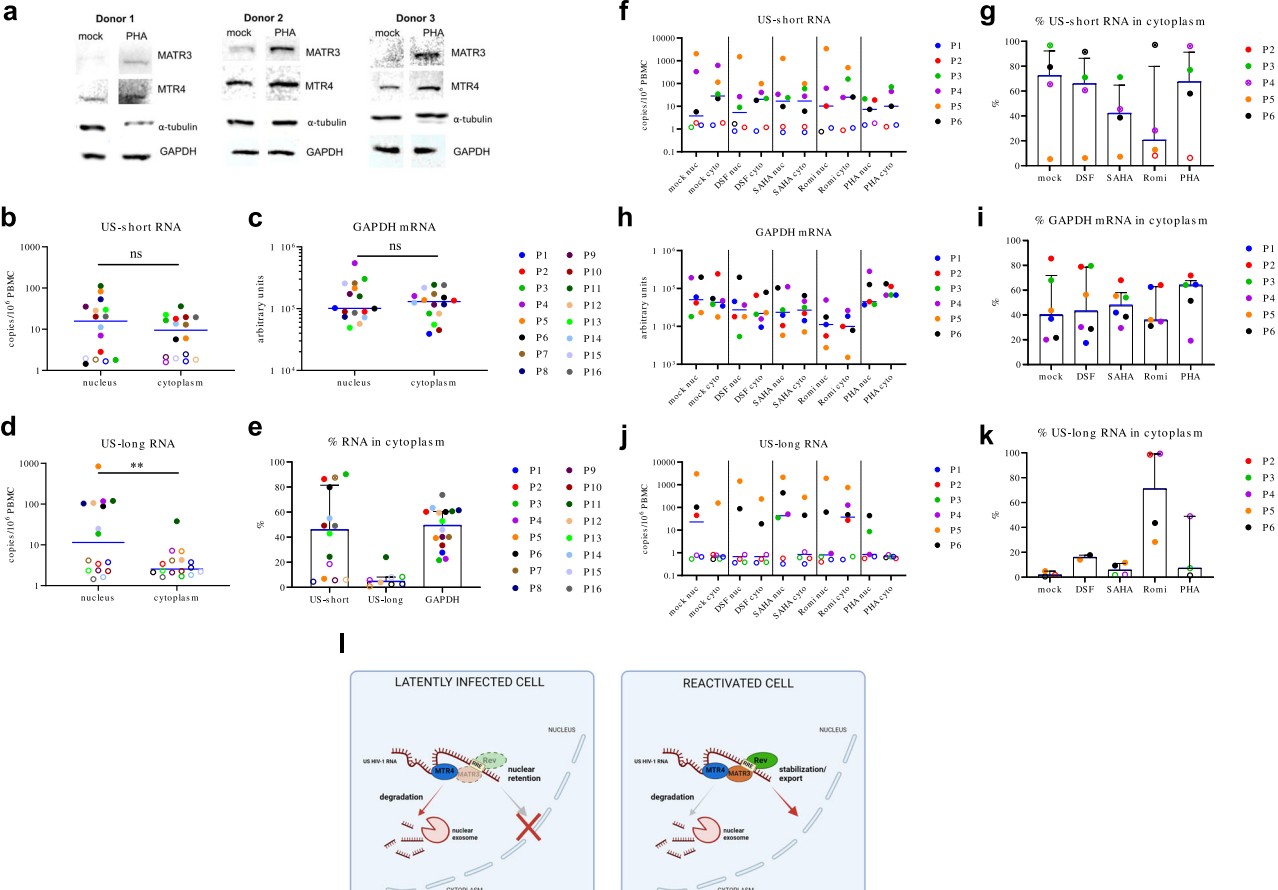

**Fig. 4 | Nuclear retention of US HIV-1 RNA in PBMCs from 22 ART-treated PWH. a** PBMCs from healthy donors were left untreated or were treated with PHA [1.66 μg/ml] and after 24 h collected for immunoblotting with the use of anti-MATR3, anti-MTR4, anti-αtubulin and anti-GAPDH antibodies. αtubulin and GAPDH are loading controls. **b**–**e** Freshly isolated PBMCs from 16 ART-treated PWH were subjected to nucleocytoplasmic fractionation for subsequent nucleic acid isolation and RT-qPCR quantification of (**b**) US-short HIV-1 RNA, (**c**) *gapdh* mRNA, and (**d**) US-long HIV-1 RNA. Percentages of each of the measured RNAs in the cytoplasmic fraction are shown in (**e**). **f**–**k** Ex vivo cultures of CD8+-depleted PBMCs from 6 ART-treated PWH were mock-treated, treated with SAHA [0.5 μM], disulfiram [0.5 μM], romidepsin [17.5 nM] or PHA [1.66 μg/ml] as a positive control in the presence of ARV [280 nM ritonavir, 180 nM azidothymidine, 200 nM raltegravir, 100 nM efavirenz]. Three days post-treatment cells were subjected to nucleocytoplasmic fractionation for subsequent nucleic acid isolation and RT-qPCR quantification of (**f**, **g**) US-short HIV-1 RNA, (**h**, **i**) *gapdh* mRNA, and (**j**, **k**) US-long HIV-1 RNA. Percentages of each of the measured RNAs in the cytoplasmic fraction are shown in (**g**, **i**, **k**). **b**–**k** Medians are indicated as horizontal lines, and for (**e**, **g**, **i**, **k**) interquartile ranges are indicated as well. The RNA copy numbers were normalized to the cell numbers measured by the beta-actin qPCR assay. Open symbols in (**b**, **d**, **f**, **j**) indicate undetectable measurements of US RNA that were assigned the values

corresponding to 50% of the corresponding assay detection limits. The detection limits depended on the amounts of the normalizer (input cellular DNA) and, therefore, differed between samples. Open symbols in **e**, **g**, **k** indicate samples where US RNA in the cytoplasmic fraction was undetectable and censored to 50% of the detection limits; because US RNA in the nuclear fraction in these samples was detectable, these circles depict the upper limits of the percentages of US RNA in the cytoplasm. Open crossed circles in **e**, **g**, **k** indicate samples where US RNA was undetectable in the nuclear fraction and censored to 50% of the detection limits; because US RNA in the cytoplasmic fraction in these samples was detectable, these circles depict the lower limits of the percentages of US RNA in the cytoplasm. Percentages of US RNA in the cytoplasmic fraction are not shown if US RNA was undetectable in both nuclear and cytoplasmic fractions. **b**–**d** RNA levels were compared between nuclear and cytoplasmic fractions using a two-sided Wilcoxon signed-rank test. **, $0.001 < p < 0.01$. ns, not significant. **l** A schematic view of the post-transcriptional regulation of US HIV-1 RNA by MATR3, MTR4, and Rev. Limiting levels of MATR3 and Rev cause nuclear retention of US HIV-1 RNA that is subjected to MTR4 for further degradation (left panel). During reactivation, MATR3 competes with MTR4 to stabilize the RNA. Rev hijacks the pathway to export the RNA (right panel). Created in BioRender. Wadas, J. (2025) https://BioRender.com/w90s107. Source data are provided as a Source Data file.

efficient discrimination between degradation or export pathways for HIV-1 RNA. In line with that, Rev was recently shown to compete with transcription/export (TREX) canonical export factor over binding to cap-binding complex (CBC) present on 5' capped US HIV-1 RNA[59], an important landmark for pre-mRNA processing including RNA export[60]. This competition between Rev and TREX is believed to suppress the export of US HIV-1 RNA via canonical TREX-dependent export and favors alternative CRM-1 dependent route[44,59]. Furthermore, a competition mechanism between MTR4 and the mRNA export adaptor ALYREF for associating with the CBC has also been demonstrated[61]. The role of CBC on 5'capped US HIV-1 RNA remains to be elucidated. However, mechanistic details about the formation of the MATR3/MTR4/Rev ribonucleoprotein complex and MATR3-MTR4 opposing activities observed here are beyond the scope of this study and will be investigated in detail in the future. Understanding how this "switch" is regulated could provide insights into the mechanisms of viral RNA stability and degradation and be critical for the design of strategies to eliminate the latent virus.

The opposing nature of the interaction between MATR3 and MTR4 with viral RNA triggered by Rev either stabilizing or degrading viral transcripts may shed light on post-transcriptional mechanisms that could be crucial for maintaining HIV-1 latency or enabling viral reactivation. We confirmed the previous finding that MATR3 levels are limited in unstimulated PBL[43] and additionally demonstrated that MTR4 levels were abundant. Notably, a recent elegant study by the Boritz group that used the FIND-seq approach directly demonstrated higher levels of several post-transcriptional factors such as those involved in mRNA degradation[53] in HIV-1 latently infected cells, highlighting that these cells have the capacity for post-transcriptional HIV-1 silencing[53]. Indeed, we report here nuclear retention of US HIV-1 RNA in PBMCs from 22 ART-suppressed PWH, suggesting a block to Rev-dependent export that may result from differential levels of MATR3 and MTR4 and/or insufficient Rev levels. The pioneering study of Pomerantz et al. has demonstrated an aberrant pattern of viral RNA expression in latently infected cell lines that was suggested to result from limiting levels of Rev[62]. Indeed, several recent studies have shown limiting levels of MS HIV-1 RNA (from which Rev is expressed) in both untreated and ART-treated infections and that it is challenging to detect MS RNA on long-term ART[51–53] that could result from insufficient splicing[63,64]. In line with these observations, we also could detect only limiting amounts of MS HIV-1 RNA in ex vivo cultures from PWH on ART. Limited levels of MATR3 (and Rev) but not MTR4 may potentially cause retention of US-long HIV-1 RNA for subsequent degradation in the nucleus in PWH cells, however direct studies will be needed to refine our model.

Length of HIV-1 US transcripts seems also to be important for nuclear export, as we did not observe any nuclear retention of short US transcripts, in contrast to long US transcripts. Indeed, as demonstrated by the Yukl laboratory, a gradient of unspliced HIV-1 transcript abundance is observed in CD4+ T cells from PWH on ART due to the constant termination of transcription during elongation[33,34]. It is possible that short transcripts are exported to the cytoplasm in an alternative, Rev-independent, fashion. Further studies are needed to understand the mechanisms involved and the role of such short viral RNA in latently infected PWH cells. Interestingly, a recent study from the Luban laboratory has demonstrated that intron-containing RNA from the HIV-1 provirus activates innate immune signaling pathways[49]. In future studies, it will be important to investigate whether the length of such transcripts determines inflammation.

Finally, we showed that the block to export of HIV-1 US-long RNA could be reversed in PWH cells using romidepsin but not using weak LRAs such as SAHA and disulfiram. This highlights the dual, transcriptional, and post-transcriptional modes of action of romidepsin. In previous ex vivo studies, romidepsin has been shown to be the most potent HDACi and to induce not only viral transcription but also

increased levels of extracellular HIV-1 RNA, and virions release from CD4+ T-cell cultures[65].

A limitation of our study is that, although we were able to demonstrate evidence of the roles of MATR3 and MTR4 in the post-transcriptional regulation of US HIV-1 RNA in co-transfection experiments, in in vitro J-Lat 9.2 cell line model of latency and in primary cell model of HIV-1 infection, their direct roles in post-transcriptional latency and reactivation in primary cell models of latency or PWH cells remain to be established. Another limitation is the exclusive focus on peripheral blood in PWH. HIV-1 persistence might be different in gut tissue vs. peripheral blood, as the former was reported to impose a stronger latency block to transcription initiation and a less strong block to elongation, resulting in lower absolute and relative (per pro-virus) HIV-1 transcription levels in rectal biopsies compared to PBMCs[34]. Further studies should reveal the differences between peripheral blood and tissues in the mechanisms of nuclear retention of HIV-1 RNA.

Altogether, this study highlights the importance of the post-transcriptional block to export as one additional mechanism leading to HIV-1 latency in PWH cells and reinforces the need to further investigate the pathways leading to full virus reactivation to hopefully reach a cure.

## Methods
### Cell lines and cell culture
J-Lat 9.2 cells are Jurkat T cells latently infected with HIV-1 strain R7/E-/GFP[66]. The Jurkat (ARP-177) and J-Lat 9.2 (ARP-9848) cell lines were obtained from the AIDS Research and Reference Reagent Program (NIAID, NIH). Cells were maintained in Roswell Park Memorial Institute medium (RPMI 1640; Sartorius #01-100-1 A) with 10% fetal bovine serum (FBS; Gibco #A5256801), and penicillin/streptomycin (PAN Biotech #P06-07100) and were cultivated at 37 °C in a 5% $CO_2$ atmosphere. Human embryonic kidney epithelial-like cells HEK 293 T (ATCC #CRL-3216) and LentiX™ 293 T (Takara Cat# 632180) were maintained in Dulbecco's Modified Eagle's medium (DMEM; Gibco #11965-092) supplemented with 5% FBS and penicillin/streptomycin.

### Reagents and antibodies
TNFα was purchased from PeproTech (#300-01 A). ABX464 (#S0076) and Ritonavir (#S118504) were purchased from Selleckchem. Phyto-hemagglutinin (PHA) (#1668) and SAHA (#SML0061) were purchased from Sigma-Aldrich. Disulfiram (#T0054) and romidepsin (#T6006) were purchased from TargetMol. Azidothymidine (#ARP-3485), ralte-gravir (#HRP-11680), efavirenz (#HRP-4624) were purchased from NIH HIV Reagent Program. Antibodies for Western blot: against MATR3 were purchased from Bethyl (1:2000, #A300-591A, rabbit) or Sigma (1:1000, MABN1587, mouse); MTR4 (1:500, #A300-614A) were purchased from Bethyl; PSF (1:4000; #P2860) and FLAG (1:1000; #F1804) were purchased from Sigma; EXOSC-10 (1:1000; #ab50558) were purchased from Abcam; GFP (1:1000, #2555), GAPDH (1:4000, #2118) and H3 histone (1:5000; #9715) were purchased from Cell Signaling Technology; α-tubulin (1:200; #sc12462-R) were purchased from Santa Cruz. Antibodies for IP and RIP: against MATR3 (#A300-591A, 2 µg) and MTR4 (#A300-614A, 3 µg) were purchased from Bethyl; GFP (#SAB4200681, 5 µg) were purchased from Sigma; FLAG (#A2220) were purchased from Millipore; rabbit IgG were purchased from GeneTex (#GTX35035, 3 µg) and mouse IgG were purchased from Millipore (#CS200621, 5 µg). Antibodies for immunofluorescence: against MATR3 (1:200, #MABN1587) and PSF (1:200; #P2860) were purchased from Sigma; MTR4 (1:200, #A300-614A) were purchased from Bethyl.

### Primary cell model for HIV-1
CD4+ T cells were isolated from buffy coats of healthy donors, obtained from the Belgian Red Cross, by negative magnetic bead

selection according to the manufacturer instructions (StemCell #19052). Following isolation, cells were stimulated with con-canavalin A (Medchemexpress #HY-P2149) for three days. Then cells were spinoculated for two hours at 800 × g at 32 °C using, per million of CD4[+] T cells, 300 ng of p24[Gag] of either VSV-G pseudotyped pNL4.3ΔRev HIV particles (Addgene #101348) or VSV-G pseudotyped wt pNL4.3 HIV particles. After spinoculation, cells were washed and cultured in a cRPMI supplemented with 5 µM of saquinavir (Medchemexpress #HY-17007) and cytokines favouring survival and quiescence. After three days, cells were harvested and used for further application.

### Infection in Jurkat T cells
Jurkat T were spinoculated for 2 hours at 800 × g at 32 °C using, per million of cells, 100 ng of p24Gag of either VSV-G pseudotyped pNL4.3ΔRev HIV particles (Addgene #101348) or wt pNL4.3 HIV particles. After spinoculation, cells were washed and cultured in a cRPMI supplemented with 5 µM of saquinavir (Medchemexpress #HY-17007) for three days, after which cells were harvested and used for further application.

### Peripheral blood mononuclear cells (PBMCs) from healthy and ART-treated PWH
PBMCs were kindly gifted from Aleksander Grabiec (Jagiellonian University, Krakow, Poland) and obtained from three healthy donors (Red Cross, Poland). PBMCs from healthy donors were seeded in a concentration of two million cells/ml in RPMI (Sartorius #01-100-1 A) supplemented with 10% FBS (Biowest #S181H-500) and penicillin/streptomycin (PAN Biotech #P06-07100). PBMCs from 16 cART-treated PWH provided by the Department of Infectious Diseases at Jagiellonian University Medical College were isolated from 5 ml of fresh blood by centrifugation on SepMate-50 (StemCell #85450) with Lymphoprep (StemCell #07851/07861). Freshly isolated PBMCs from ART-treated PLWH were subsequently subjected to nuclear and cyto-plasmic fractionation protocol.

### Lentiviral production and transduction
VSV-G-pseudotyped lentiviral particles were obtained by PEI transfection of HEK293T with plasmids pLKO.1 shMATR3-905 or pLKO.1 shLUC[43], co-transfected with pSPAX2 and pMD2G plasmids. After 16 h, the DMEM medium was replaced for RPMI, and lentiviral particles were collected 48 and 72 h after transfection, filtered through 0.22 µm syringe filter (TPP #99722), and concentrated 40× using Amicon® Ultra-15 Centrifugal Filter Unit 10 kDa MWCO (Merck Millipore #UFC901024), aliquoted and stored at −150 °C until further use. The concentration of lentiviral particles was measured by HIV-1 p24 ELISA (Xpress Bio #XB-1000) according to manufacturer protocol. Lentiviral transduction of J-Lat 9.2 cells was performed by spinoculation (800 × g, 90 min, 32 °C) using 10 µg of lentiviral particles (MOI 10) per one million of cells in the presence of polybrene (8 µg/ml) in total 100 µl of RPMI. After spinoculation, cells were maintained at a concentration of one million cells/ml in RPMI. After 24 h, cells were diluted three times in RPMI, and puromycin (BioShop #PUR555.2) was added to the concentration of 1 µg/ml. Cells were cultured for 72 h for further applications.

### Flow cytometry
J-Lat 9.2 were collected 24 h after stimulation, centrifuged, washed in PBS, and resuspended in 3.7% paraformaldehyde (PFA; Sigma #P6148) in PBS for fixing. After 30 min, cells were washed twice in PBS, and the percentage of GFP-positive cells was analyzed using BD LSR Fortessa.

### Cellular viability test
Cellular viability was assessed using colorimetric XTT assay (Biological industries #20-300-1000) according to manufacturer's instructions.

Absorbance was measured using microplate reader (SpectraMax iD5, Molecular Devices)

### RNA FISH, immunofluorescence, and confocal imaging
J-Lat 9.2 were collected 24 h after stimulation, centrifuged, and immobilized on poly-L-lysine-coated coverslips by incubation through 1 h in RPMI supplemented with 50% FBS. Next, cells were washed with PBS and fixed for 30 min in 3.7% PFA buffered with PHEM, and per-meabilized for 10 min in PBS with 0.1% Tween 20. Then, cells were subjected to RNA FISH protocol (Molecular Instruments; HCR™ RNA FISH), as described in ref. 67 with the use of the set of 20 probes designed by Molecular Instruments for the HIV-1 *gag* region. More specifically, the cells on coverslips were incubated with 30% Probe Hybridization Buffer (Molecular Instruments, Inc. #BPH03821) for 30 min at 37 °C, then with the HIV-1 *gag* probes (1:250) in the 30% Probe Hybridization Buffer, in 37 °C, overnight. Subsequently, samples were washed in the warm 30% Wash Buffer (Molecular Instruments, Inc. #BPW01522) four times and in 5× SSCT (0.75 M NaCl, 75 mM sodium citrate pH 7.0, 0.1% Tween 20) buffer twice. Next, the cover-slips were incubated in the Amplification Buffer (Molecular Instruments, Inc. #BAM01522) for 30 min, room temperature, and at this time, the amplifiers B1-h1 (Molecular Instruments, Inc. #S013922) and B1-h2 (Molecular Instruments, Inc. #S012522) conjugated with Alexa Fluor 647 were prepared by the snap-cooling. The amplification step was performed by incubation of coverslips with amplifiers (1:50) in the Amplification Buffer at room temperature overnight. After amplification, the coverslips were washed five times in 5× SSCT and twice in PBS. Next, samples were blocked in 4% Bovine Serum Albumin in 0.1% PBS-Tween 20 and subjected to immunofluorescence with the use of indicated antibodies diluted in 1% BSA in 0.1% PBS-Tween 20: anti-Lamin β1 (Abcam #ab16048, 1:500), anti-MATR3 (Sigma #MABN1587, 1:200), anti-MTR4 (Thermo Scientific A300-614A#, 1:200). Samples were incubated with the primary antibodies in 4 °C, overnight. Secondary antibodies were diluted 1:400 in 1% BSA in 0.1% PBS-Tween 20: anti-rabbit Alexa Fluor 594 (Thermo Scientific #A11012), anti-mouse Alexa Fluor 546 (Thermo Scientific #A10036), and incubated with coverslips through 2 h at room temperature. Nuclei were stained using DAPI (4',6-diamidino-2-phenylindole dihydrochloride; Thermo Scientific #D1306) and the slides were mounted with the use of Prolong Diamond Antifade Mounting Medium (Invitrogen #P36970). Images and Z-stacks were acquired by confocal microscope ZEISS LSM 880 with 100×/1.46 NA Plan Apochromat Objective with oil immersion and the ZEN Imaging Software (ZEISS), and analyzed using Fiji software[68]. To quantify the number and volume of spots, after being imported via the Bio-Formats plugin[69], the images were segmented, and 3D structures larger than 5 voxels were subsequently identified using the 3D Object Counter plugin[70]. The overlap (for volume or number) was computed by dividing the shared signal (CH1&2) by the maximum value between the two channels individually according to the following formula CH1&2/(max(CH1,CH2))*100%.

### RNA stability test
Transduced J-Lat 9.2 cells were centrifuged 24 h after stimulation, and samples at time 0 min were washed in PBS and resuspended in TRIzol Reagent (Thermo #15596018). The remaining cells were resuspended in RPMI medium containing 10 µg/ml actinomycin D (Abcam #ab141058) or DMSO (Sigma #D4540), maintained for 30 and 60 min, washed in PBS, and resuspended in TRIzol for further RNA isolation according to manufacturer protocol.

### siRNA-mediated knock-down and co-transfection in HEK 293 T cells
siRNA targeting MTR4, and non-targeting control were obtained from Dharmacon (ON-Target plus Smart Pool SKIV2L2 #L-031902-02-0005; ON-Target plus Control Pool #D-001810-10-05). HEK293T cells seeded

on coverslips were transfected with 10 nM of siRNA using Lipofectamine RNAiMAX (Thermo #13778-075), according to manufacturer protocol, and after 24 h, medium was changed, and the cells were transfected again with 10 nM of siRNA. The medium was changed after 24 h, and the cells were transfected with the plasmids: pHIV-INTRO, pTat-101, pRev-EGFP, or pCDNA3.1, using Lipofectamine 2000 (Thermo #11668-019), according to manufacturer protocol. 24 h post-transfection, HEK293T cells were collected for further applications.

## Immunoprecipitation and benzonase treatment

For IP against FLAG-MS2nls, HEK293T cells were transfected with plasmids pHIV-INTRO, pTat-GFP, and pFlag-MS2nls as described previously[41]. 24 h post-transfection, cells were scraped and lysed in 1 ml of RIPA Lysis buffer (50 mM TRIS-HCL pH 7.4, 150 mM NaCl, 1% NP-40, 0.1% SDS, 1.5 mM MgCl$_2$, 1 mM PMSF, 0.1 mg/ml Dextran, 10 mM RVC, Protease inhibitor cocktail) for 30 min in 4 °C. Then, the lysates were centrifuged 15 min, 10,000 rpm at +4 °C and supernatants were transferred onto 30 μl anti-flag beads (Sigma #F1804) and incubated 4 h on rotor at +4 °C with Yeast t-RNA [100 μg/ml] (Thermo Scientific AM7119). All IPs were washed six times in RIPA wash buffer (50 mM TRIS-HCL pH 7.4, 300 mM NaCl, 1% NP-40, 0.1% SDS, 1.5 mM MgCl$_2$, 1 mM PMSF, 0.1 mg/ml dextran, 0.2 mg/ml heparin) by incubation 5 min on a rotor at +4 °C and subsequent centrifugation 5000 rpm, 1 min at 4 °C. Immunoprecipitated complexes were eluted by boiling in 2×Laemmli Sample buffer at 95 °C for 10 min. Obtained samples were then subjected to SDS-PAGE and Western blot. For IP against FLAG-MATR3 and MTR4, LentiX HEK293T cells were plated onto 15 cm plate at density of 1.5 × 10$^7$ cells per plate. On the next day cells were transfected with pHIV-INTRO, pTat-101, pRev-GFP (for IP against MTR4) or additionally transfected with either with pCMV-empty or pCMV-FLAG-MATR3 plasmids (for IP against FLAG). At 24 h post-transfection cells were collected in RNA-immunoprecipitation RIP buffer (20 mM Hepes pH 7.9, 150 mM NaCl, 1% NP-40, 0.1% Triton X-100, 2.5 mM MgCl$_2$, 1 mM PMSF, 1 nM DTT) supplemented with RiboLock RNase inhibitor (Thermo Scientific, Cat. EO0381) and DTT to final concentration of 40 U/ml and 1 mM respectively, and incubated for 30 min with rotation at 4 °C. Next, crude lysates were centrifuged at 10,000 RPM for 15 min at 4 °C, supernatants were collected and incubated for four hours with previously equilibrated Anti-FLAG® M2 Magnetic Beads (Milipore, Cat. M8823) at 4 °C with rotation. Next, beads were washed 4-times using RIP buffer followed be Benzonase (Millipore, Cat. 70664-3) treatment (75 units/IP reaction) for two hours with rotation at 4 °C with rotation. Beads were washed again 4-times using RIP buffer and immunoprecipitated complexes were eluted by boiling in 2×Laemmli Sample buffer at 95 °C for 10 min. Obtained samples were then subjected to SDS-PAGE and Western blot.

## RNA immunoprecipitation (RIP)

HEK293T cells on 100 mm petri dish were transfected with pHIV-INTRO, pTat-101, and pRev-GFP and after 24 h trypsynized, washed in cold PBS, and lysed at +4 °C for 15 min in RIP buffer (20 mM Hepes pH 7.9, 150 mM NaCl, 1% NP-40, 0.1% Triton X-100, 2.5 mM MgCl2, 1 mM PMSF, 1 nM DTT) in 1 ml per petri dish supplemented with RiboLock [160 U/ml] (Thermo Scientific #EO0382) and protease inhibitor cocktail (Roche #04693116001). 10% of lysates were saved as input samples. Anti-MTR4 (Bethyl #A300-615A) or anti-MATR3 (Bethyl #A300-591A), and Rabbit IgG isotype antibodies (GeneTex #GTX35035) were coupled to G PLUS agarose (Santa Cruz, sc-2002), by adding 2 μg of antibodies to 30 μl of beads per sample and incubated on a rotor 2 h at +4 °C. Equal amount of the lysates (in mg) was loaded onto the beads coupled with antibodies, incubated 4 h on rotor at +4 °C and washed five times in RIP buffer supplemented with RiboLock [2 U/ml] and protease inhibitor cocktail by incubation of the IPs 5 min on rotor at +4 °C, for subsequent centrifugation 5000 rpm, 1 min, +4 °C. IPs were resuspended in 100 μl of RIP buffer, 10 μl of RIPs were eluted in

Laemmli Sample Buffer for Western blot, remaining 90 μl as well as input samples were subjected to RNA isolation by TRIzol protocol.

For RIP in primary and Jurkat cells were washed with PBS and lysed in SDS lysis buffer buffer (Sigma-Aldrich #20-163) for 10 min at +4 °C. Anti-MTR4 (Bethyl #A300-615A) or anti-MATR3 (Bethyl #A300-591A), and Rabbit IgG isotype antibodies (Cell signaling##2729) were incubated overnight with equals amounts of lysate (equivalent to 10 million of cells/condition). The next day, 30 μL of protein G magnetic beads (Cell signaling#9006S) were added to the samples and incubated on a rotor 2 h at +4 °C. Samples were then washed five times in RIP buffer supplemented with with RiboLock [2 U/ml] and protease inhibitor cocktail, by incubation of the IPs 5 min on rotor at 4 °C, for subsequent incubation on a magnetic rack, 4 °C. 10 μl of RIPs were subjected to RNA isolation by TRIzol protocol. RNA was extracted also from input samples.

## Reciprocal immunoprecipitation (re-IP)

LentiX HEK 293 T cells were plated onto 15 cm plates at a density of 1.5 × 10$^7$ cells per plate. On the next day, cells were transfected with pHIV-INTRO, pTat-101, pRev-GFP, and either with pCMV-empty or pCMV-FLAG-MATR3 plasmids. At 24 h, post-transfection cells were collected in RNA-immunoprecipitation (RIP) buffer supplemented with RiboLock RNase inhibitor (Thermo Scientific, Cat. EO0381) and DTT to a final concentration of 40 U/ml and 1 mM, respectively, and incubated for 30 min with rotation at +4 °C. Next crude lysates were centrifuged at 5000 × g for 15 min at +4 °C, supernatants were collected and incubated overnight with previously equilibrated Anti-FLAG® M2 Magnetic Beads (Milipore, Cat. M8823) at +4 °C with rotation. On the next day, beads were washed 5 times using RIP buffer and subjected to FLAG elution done by two incubations with RIP buffer supplemented with 3×FLAG peptides (Sigma-Aldrich, Cat. F4799) at 150 ng/ml, 30 min each. The eluates were then divided into two equal parts by volume, adjusted to a final volume of 1 ml with RIP buffer, and subjected to the second round of immunoprecipitation with 5 μg of α-IgG (Milipore, Cat. CS200621) or 5 μg of α-GFP (Sigma-Aldrich, Cat. SAB4200681) antibodies overnight at +4 °C with rotation. On the next day, Protein G PLUS Agarose (Santa Cruz, Cat. Sc-2002) was added and incubated for 4 h at +4 °C with rotation. Beads were then washed 5 times and precipitated complexes were eluted by boiling in 2×Laemmli sample buffer at 95 °C for 10 min. Obtained samples were then subjected to SDS-PAGE and western blotting.

## Nuclear and cytoplasmic fractions

J-Lat 9.2, primary CD4$^+$ T cells, CD8$^+$-depleted PBMCs from cART-treated HIV$^+$ patients were subjected to nuclear and cytoplasmic fractionation protocol as described previously[41]. More specifically, 5 × 10$^6$ J-lat 9.2 cells, 8 × 10$^6$ CD4$^+$ T cells, 6 × 10$^6$ CD8$^+$-depleted PBMCs were centrifuged, washed in cold PBS, and resuspended in 200 μl of a cold hypotonic buffer A (20 mM TRIS-HCl pH 7.5, 10 mM NaCl, 3 mM MgCl$_2$, 10% glycerol, prepared in RNase-free water) with the protease inhibitors cocktail and incubated 1 min on cooler. Next, 10% NP-40 was added to the final concentration of 0.5%, and samples were vortexed extensively, incubated for 5 min on ice, and centrifuged (9000 rpm, 5 min, +4 °C). The supernatant was collected to the new tubes as a cytoplasmic fraction, and the nuclei were washed once in Buffer A and resuspended in 200 μl of the buffer A, as a nuclear fraction. Obtained fractions were subjected to further applications.

## RNA extraction and RT-qPCR quantification

Extracellular HIV RNA was isolated using Viral RNA Kit (A&A Biotechnology, #034-200) and reverse-transcribed using High-Capacity cDNA Reverse Transcription Kit (Thermo Scientific #4368814). Obtained cDNA was quantified by TaqMan-based qPCR, using gag-p24 primers and probe and RT-PCR Mix Probe (A&A Biotechnology #2008-2000P), and expressed as the RNA copies/ml of the supernatant.

RNA from cell-associated total, nuclear, cytoplasmic, or from IP from HEK293T, J-Lat 9.2, Jurkat, and primary CD4+ T cells were isolated using TRIzol Reagent (Thermo #15596018) according to manufacturer protocol and treated with TURBO DNase (Thermo Scientific #AM2238). Then, 500 ng of RNA or total RNA form RIPs were subjected to reverse transcription using High-Capacity cDNA Reverse Transcription Kit (Thermo Scientific #4368814), and obtained cDNA was subjected SYBR green-based qPCR using either GO-Taq MM (Promega #A6002) for J-Lat 9.2 and HEK 293 T or either Luna universal qPCR master mix (New England Biolabs #M3003) with primers targeting (i) unspliced HIV-1 RNA (set primers for RRE region), (ii) multiply-spliced HIV-1 RNA (MS_Fw and MS_Rev), and (iii) total HIV-1 RNA (primers for TAR RNA). RNA levels were normalized to GAPDH mRNA (HEK 293 T samples), or 18S rRNA (J-Lat 9.2 samples), or TBP for Jurkat and Primary CD4 + T cells and shown as a fold change of relative expression compared to the control sample. RNA levels from RIPs were normalized to input and then to IgG control. Oligonucleotide sequences are listed in Supplementary Data 1.

### Study subjects
We selected 22 HIV-1-infected individuals at the Jagiellonian University Medical College (Krakow, Poland) based on the following criteria: all participants were treated with cART for at least 6 months, had an undetectable plasma HIV-1 RNA level (20 copies/ml) for at least 6 months and had a level of CD4 + T lymphocytes higher than 200 cells/mm$^3$ of blood. Characteristics (year of birth, duration of therapy, nadir CD4 + T-cell count, CD4+ nadir, antiviral regimens) of PWH from the Jagiellonian University Medical College are presented in the Supplementary Table 1.

### Ethics statement
Ethical approval (Approval No. 1072.6120.184.2018) was granted by the Human Subject Ethics Committees of the Jagiellonian University Medical College (Krakow, Poland). All individuals enrolled in the study provided written informed consent for donating blood.

### Ex vivo cultures of CD8+-depleted PBMCs from ART-treated PWH
PBMCs were isolated from 40 ml of fresh blood from 6 ART-treated PWH obtained from the Department of Infectious Diseases at Jagiellonian University Medical College (Poland). More specifically, the blood was treated with RosetteSep Human CD8 Depletion Cocktail (StemCell #15663), and subsequently cells were isolated from the buffy coat from the whole blood centrifugation on SepMate-50 (StemCell #85450) with Lymphoprep (StemCell #07851/07861) and maintained at concentration $2 \times 10^6$ cells/ml in RPMI (Sartorius #01-100-1 A) supplemented with 10% FBS (Biowest # S181H-500) and penicillin/streptomycin (PAN Biotech #P06-07100). Cells were seeded in a fresh RPMI medium at a concentration of $2 \times 10^6$/ml (total $6 \times 10^6$) per condition.

### Quantification of proviral HIV-1 DNA
Samples from infected Jurkat or primary CD4 + T cells were used to quantify HIV-1 total DNA as previously described[71].

### RNA/DNA isolation and quantification from PBMCs
Total nucleic acids were extracted from nuclear and cytoplasmic fractions of cART-treated PWH-derived PBMCs using Boom isolation method[72]. Extracted RNA was treated with DNase (TURBO DNA-free Kit, Thermo Scientific #AM1907) to remove genomic DNA that could interfere with quantitation. DNase-treated RNA was subjected to a onestep RT-PCR for HIV-1 US-short RNA, HIV-1 US-long RNA, and HIV-1 MS RNA that was performed using QIAGEN OneStep RT-PCR Kit (Qiagen #210215) with the following primer pairs: M_rev + Ψ_F, GAG1 + SK431, and MS_total + mf83, respectively. Two-microliter aliquots of this RT-PCR were subsequently used as input for seminested US-short, US-

long, and MS RNA qPCR in which the following primer/probe combinations were used: Ψ_F + Ψ_R + Ψ_Probe, GAG1 + GAG2 + GAG3, and mf84 + mf83 + ks2-tq, respectively. For quantitation of cellular *gapdh* RNA, an aliquot of DNase-treated RNA was reverse-transcribed using random primers (Thermo Scientific #48190011) and SuperScript III reverse transcriptase (Thermo Fisher Scientific #18080093). cDNA was then subjected to qPCR using a TaqMan Gene Expression Assay for GAPDH (ID: Hs02758991_g1, Thermo Fisher Scientific #4331182). β-actin DNA was measured by qPCR using TaqMan™ β-Actin Detection Reagents (Thermo Scientific # 401846).

### Statistics and reproducibility
For each examined condition, at least three independent biological replicates in duplicate were performed, as indicated in the figure legends. The representative experiments shown in Fig. 2a–d were repeated at least twice. Mean values are shown with the standard error of the mean (SEM). RNA levels were compared between nuclear and cytoplasmic fractions in the PWH samples using a Wilcoxon signed-rank test. Statistics were performed using a Wilcoxon signed-rank test for HIV integration and a two-way ANOVA test for the comparisons in a primary cell model of infection. In in vitro studies statistical significance was measured with a Student's *t* test. Significant *p* values are indicated by the asterisks above the graphs ($p \le 0.05$ [*], $p \le 0.01$ [**], $\le 0.001$ [***], $\le 0.0001$ [****]). Analyses were performed using Prism version 6.0.

### Reporting summary
Further information on research design is available in the Nature Portfolio Reporting Summary linked to this article.

## Data availability
Data underlying Figs. 1a–i, 2a–d, 2f–i, 3a–h, 4a–k, Supplementary Figs. 2a, b, 3a–h, 4a–d, 5a–f are provided as Source Data files. Source data are provided with this paper.

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

## Acknowledgements

We thank the HIV+ individuals for their willingness to participate in this study. We thank the nursing team at University Hospital in Krakow who cared for the HIV+ individuals. We thank Claire Thiry from the Transfusion Center of Charleroi (Belgium) for providing blood from healthy donors. We thank IFOM for providing access to the server. We thank Danuta Earnshaw and Magda Masłoń (Małopolska Centre of Biotechnology, Jagiellonian University, Kraków) for their critical review of the manuscript. A.K.-P., A.Dorman, and H.A. acknowledge funding from the National Science Centre, Poland (Sonata Bis Grant UMO2018/30/E/NZ1/00874). A.K-P. and J.W. acknowledge funding from the National Science Centre, Poland (OPUS Grant UMO-2022/45/B/NZ3/03890). C.V.L. acknowledges funding from the Belgian National Fund for Scientific Research (F.R.S-FNRS, Belgium), the French INSERM agency "ANRS/Maladies infectieuses émergentes", ViiV Healthcare, the "Fondation Roi Baudouin", the Internationale Brachet Stiftung (IBS), The "Amis des Instituts Pasteur à Bruxelles, asbl", and the US National Institutes of Health (NIH) (MDC grant UM1AI164562 co-funded by National Heart, Lung and Blood Institute, National Institute of Diabetes and Digestive and Kidney Diseases, National Institute of Neurological Disorders and Stroke, National Institute on Drug Abuse and the National Institute of Allergy and Infectious Diseases). M.B. was funded by fellowships from the Belgian « Fonds pour la formation à la Recherche dans l'Industrie et dans l'Agriculture (FRIA) (F.R.S.-FNRS) » and then from "Les Amis des Instituts Pasteur à Bruxelles, asbl". A. Dutilleul was funded by an "Aspirant" fellowship from the F.R.S.-FNRS, by a fellowship from the "Les Amis des Instituts Pasteur à Bruxelles, asbl", and then by a "PDR" grant (PDR 40021157) from the F.R.S-FNRS. C.V.L. is "Directrice de Recherches" of the F.R.S-FNRS. The laboratory of C.V.L. is part of the ULB-Cancer Research Centre (U-CRC) (Faculty of Medicine, ULB). Work in the A.O.P. laboratory is supported by grants from the Dutch Medical Research Council (ZonMw) (09120011910035), from Gilead Sciences Research Program (10904), and from TKI-PPP grants Target2Cure by Health Holland/Aids Fund (LSHM19101-SGF) and Innovation Exchange Amsterdam (2019-1167). The authors would like to thank IFOM for providing access to the server. P.M. is supported by the Italian Association for Cancer Research (AIRC), Investigator Grants (#24976), and by the Ministero dell'università e della ricerca (MUR), PRIN PNRR 2022 (P2022F3YRF). The open-access publication of this article was funded by the Priority Research Area BioS under the program "Excellence Initiative – Research University" at the Jagiellonian University in Krakow.

## Author contributions

Led the study: A.K.-P. Conceived and designed the study: A.K.-P., C.V.L., A.O.P. Supervised the work: A.K.-P., C.V.L., and A.O.P. Conceptualized, planned, and designed the experiments: A.K-P., C.V.L., A.O.P., A.Dorman, M.B. Performed most of the experiments and prepared the figures: A.Dorman and M.B. Led quantifications in PWH cells: A.O.P. Performed quantifications in PWH cells: A.O.P., A.V., M.B., L.N., A.Dorman, V.A-F., G.M., C.N., S.D.W. Performed re-IP: J.W. Led the experiments in primary cell models: C.V.L. Performed the experiments in primary cell models: M.B., L.N., and A.Dutilleul. Performed HIV+ individuals' selection: M.B.-J. Confocal microscopy data quantification: P.M. Analyzed and interpreted the data: A.Dorman, M.B., A.V., J.W., H.A., P.M., G.M., K.P., A.O.P., C.V.L., and A.K.-P. Contributed with reagents/materials/analysis tools: A.K.-P., C.V.L., A.O.P., A.M., P.M., K.P., V.A-F. Wrote the manuscript: A.K.-P. and A.Dorman. Acquired funding: A.K.-P., C.V.L. All authors read or provided comments on the manuscript. All data were validated by A.K.-P., C.V.L., A.O.P., A.Dorman and M.B. A.V., and J.W. equally contributed to this work.

## Competing interests

A.O.P. received a research grant from Gilead Sciences Research Program. C.V.L. received a research grant from ViiV Healthcare. The funders had no role in study design, data collection and analysis, decision to publish, or preparation of the manuscript. The remaining authors declare no competing interests.
