## [Transparent Peer Review file · Nature Communications]

Nuclear retention of unspliced HIV-1 RNA as a reversible post-transcriptional block in latency.

Corresponding Author: Professor Anna Kula-Pacurar

Version 0:

Reviewer comments:

Reviewer #1

(Remarks to the Author)

The manuscript, "Nuclear retention of unspliced HIV-1 RNA as a novel reversible post-transcriptional block in latency" by Agnieszka Suder et al. attempts to characterize the activity of a small molecule Rev inhibitor ABX-464 in a contribution to HIV-1 latency. The authors use a J-Lat9.2 cell line harbouring an intact HIV-1 genome with multiply spliced HIV-1 RNAs detectable by GFP fluorescence. Using various manipulations including siRNA-mediated depletion, they implicate a relationship between related host factors MATR 3 & 4, a previously described phenomenon found to be important for nucleocytoplasmic export of RRE-containing RNAs. In general, the manuscript describes novelty as there is very little available information about such a post-transcriptional block in latent HIV-1 infections. Moreover, little is known about the involvement of the exosome in maintaining a pool of HIV-1 RNAs in latently infected cells.

This reviewer would offer a few required suggestions on how to convincingly show involvement of the MATR3 and 4 axis on RNA nuclear retention in latently infected cells.

1. The images shown in Supp Figure 2 are not convincing. Although T cells are generally difficult to image considering their large nuclei, the authors did not present control images. Moreover, a focal plane of many cells in culture should be presented for a more convincing presentation.
2. The authors employ qRT-PCR in many of the figures reflecting nuclear vs cytoplasmic preparations. Something is missing to instill confidence in these analyses. qRT-PCR is sensitive, but perhaps a bit too sensitive. Northern blotting could be performed to echo these data. The authors might expect the reader to take a leap of faith in accepting the data as written; a more convincing representation of the data is warranted for Nat. Comms.
3. The data with the 12 PLWH are quite skewed and are confusing to this reviewer. The authors should use these primary cells and examine the distributions of viral RNAs by imaging analyses.
4. The siRNA knockdown western blot is not convincing. A mere two-lane blot demonstration as presented is rather elementary at this stage (Figure 1D).
5. Legends are incomplete and unclear. E.g., legend text for Figure 4D is missing and the title of Figure 3 is missing something. This reviewer suggests to carefully edit the legends for completeness.
6. Data presented in Figure 6 on nuclear retention of RNAs are interesting, but the authors should necessarily isolate relevant HIV-1-tropic cell types from pooled PBMCs for this analysis.
7. The Discussion is interesting too, but it reads a bit like a review and is not interpretative of the data shown in the manuscript.

Reviewer #2

(Remarks to the Author)

Suder et al. have investigated post-transcriptional blocks to reactivation from latency of HIV-1. Indeed, novel strategies, including the targeting of post-integration, post-transcriptional processes, are valuable additions to anti-HIV therapy. In this study, they have focused on the export of unspliced HIV-1 RNAs, which is dependent of the viral protein Rev. They suggest a model in which Rev and a known partner, Matr3 (MATR3), favor the export of unspliced RNAs containing the rev-responsive element (RRE) in reactivated cells, whereas in latently infected cells in which neither MATR3 or Rev are highly expressed, the RNAs are bound by MTR4 helicase and targeted to the nuclear exosome for degradation. However, several

aspects of the model have not been clearly demonstrated. This reviewer also feels that the organization of the manuscript is not optimal. The links between the different sections and how they relate to the overall model is not always clear.

1. To address the importance of Rev in viral reactivation, the authors have relied on the use of a drug, ABX464. While initially described as an inhibitor of Rev, its mechanism of action remains unclear. It has not been shown to bind Rev, but instead binds the Cap-Binding-Complex. Even so, the mechanism by which ABX464 affects the export of unspliced HIV RNAs is fairly controversial. Recent studies clearly show that ABX464 upregulates miR124 in immune cells and induces a strong anti-inflammatory effect. The drug is being trialled for treatment of SARS-Cov2 infections for example. Furthermore, a recent study confirmed that ABX464, known clinically as Obefazimod, reduced the HIV reservoir, upregulated mir124 and reduced chronic inflammation in HIV patients. Thus, it seems unlikely that ABX464 is a specific inhibitor of Rev, and importantly for the present study, may very likely affect the immune activation of the cells, which is well known to be important for HIV-1 infection. Thus, it's difficult to interpret the results using this drug shown in Figures 1, 2 and 7, in terms of a specific effect on Rev.

The authors found that RRE-containing RNA in the nucleus is unstable in the absence of Rev partner, MATR3. They suggest an overall model in which MATR3 competes with MTR4, in which MATR3 promotes rev-dependent export or MTR4 promotes transcript degradation by targeting to nuclear exosome. However, several aspects of this model need to be confirmed. First, MATR3 and MTR4 would found to interact with HIV-1 RNA using an MS2 pull-down assay (Fig 4). This analysis should be complemented with standard RNA immunoprecipitation analysis, using antibodies to endogenous proteins.

Co-immunoprecipitation analysis showed that MATR3 interacts with MTR4 and with Rev, and that MTR4 interacts with MATR3 and with Rev. The authors conclude that all 3 proteins form a complex. However, these analyses do not show that all 3 factors are found in the same complex. The proteins could be in separate complexes (for example MATR3 and MTR4 and MATR3 and Rev), especially as the MATR3-MTR4 interaction appears quite weak, whereas MTR4 seems to interact very well with Rev. The authors should perform re-IPs to address whether all 3 proteins are found in the same complex. Indeed, according to the final model, all 3 proteins would not be expected to be found together on HIV RNA. Additionally, it is important to address whether these interactions are dependent on RNA by performing the IPs in the presence and absence of RNase.

They next addressed the co-localization of MTR4, MATR3 and HIV-1 RNA by immune RNA-FISH. The results are not compelling. There does not seem to be a particularly significant overlap of the 3 molecules. However, it would be necessary to perform quantification and statistical analysis to test this. Again, given that the overall model suggests that these factors may compete for the RNA, would an overlap be expected?

In Figure 5, the authors depleted MTR4 and found that the abundance of Rev-dependent transcripts increased, and more transcripts were associated with MATR3, leading to the notion that MTR4 and MATR3 exhibit antagonizing activities. An alternative explanation could be that loss of MTR4 leads to accumulation of HIV RNA, and over-expression of Rev, together with its partner MATR3, allows the transcript to be exported to the cytoplasm where it is protected from degradation. In this case, the factors are not necessarily antagonistic, but rather Rev/MATR3 exports the RNA that has been stabilized by loss of MTR4. If the authors wish to claim antagonistic activities of MTR4 and MATR3, additional experiments that directly address the antagonism must be provided.

These sections (figures 4 and 5) could be reorganized to provide a clearer idea of how the authors interpret their data. In figure 4, they claim that the factors form a complex, but in figure 5, they claim they have antagonistic activities. The authors should reorganize or better explain the interpretation of the data to provide a clearer flow of their logic for the reader.

The data shown in Figure 6 is problematic for several reasons. In the experiments using 6 patients, the analysis sometimes ends up using 2 or 3 samples, which are not always the same set analysed between the various treatment conditions, and which are quite variable amongst themselves. There is no statistical analysis shown and it may even be quite difficult to perform statistics on these results. Also, it isn't clear what the conclusion of these experiments really is, and how it relates to the previous figures. This is another example of a suboptimal organization of the manuscript. Furthermore, what is the short unspliced RNA in the cytoplasm?

In fig 7, the use of ABX464 to demonstrate that MTR4 hampers viral reactivation via Rev complicates the interpretation of the experiment. In the presence of this anti-inflammatory drug, a higher proportion of HIV-1 transcripts may be aberrant, which would explain the increased association with MTR4. The authors should use an alternative strategy to demonstrate a dependence on Rev.

Additional comments:

Many of the western blots are of poor quality. It would be preferable to have better quality blots to increase the robustness of the data.

Additional controls for nuclear and cytoplasmic fractions should be provided. For RNA analysis, it is more appropriate to quantify a nuclear RNA, such as 7SK or an snRNA. Quantifying DNA is not sufficient to assess contamination of the cytoplasmic fraction by the nucleoplasm. For immunoblotting analysis, a nucleoplasmic factor, such as a transcription factor or a splicing factor, should be included. Histones mark the chromatin fraction.

Reviewer #3

(Remarks to the Author)

In this study, Suder et al. study blocks to HIV RNA nuclear export in the context of viral latency reversal, focused largely on the effects of the drug ABX-464 that is known to reduce Rev activity and a host factor, Matr3, implicated as a positive regulator of Rev.

To me, this is interesting work but it feels like two developing stories. In the first part of the paper (Figures 1 and 2) the authors show that ABX-464 or MATR3 shRNA each reduce virus output during attempts to reverse HIV latency in a JLAT 9.2 model. However, based on the literature blocking Rev should block viral output in any setting so that the significance of these results is not very clear. Later (Figure 7), MATR3 levels were shown to be upregulated by LRAs in human donor cells (Figure 7A) but the authors did not attempt to correlate these effects (or, alternatively, whatever ABX-464 is targeting) to their patient cell post-transcriptional block.

The second part of the paper (Figures 3-5) is focused more on mechanism and addresses the role of MATR3 in regulating HIV RNA stability, suggesting a link between MATR3 and MTR4 (a regulator of the RNA exosome). I think the data are convincing that MATR3 regulates HIV RNA stability, and I also found it interesting that MATR3 looks to differ from ABX-464 in its overall mode of action. However, the argument for a direct link to MTR4 in this pathway looks premature, considering that MTR4 depletion appears to play a general role in regulating viral RNA levels whether or not they are Rev-dependent (Figure 5).

Overall, this is a broad study with interesting observations here but a weakness in cohesion. Some more specific comments are as follows.

Major:

1. Figure 2. In 2D it seems strange that a >1000-fold increase in RNA levels based on qRT-PCR doesn't translate to the FISH assay where differences are quite modest. Also effects in Figures 2 and 3 are minor compared to Figure 1. Are these assays comparable? What explains these discrepancies?
2. Related, in 2E, what is the resolution of the system that allows the authors to determine that dot volumes are greater or less than 1 μ m? Presumably the argument here is that these are not single RNAs but granules of some nature, correct? The rigor of this system and relevant caveats could be better explained. Related, in 3E if MATR3 is affecting stability then shouldn't we would expect a bigger effect on volume in this setting relative to the drug?
3. Figure 3H is hard to interpret without a TNF alone control
4. Figure 4. There are some confusing discrepancies in these experiments, e.g., 4A tracks Tat and RNA, while 4B and 4C track Rev but not Tat or the viral RNA. Also, wouldn't the most relevant experiment to look at MTR4/exosome-HIV interactions with or without MATR3?
5. Figure 5. why move to 293T cells for these experiments?
6. Figure 7B. Why move to U1 cells for these experiments?

Minor:

1. Figure 3. "Retain" may be wrong word choice if it's stability.
2. Figure 3F. Should be noted here that ActD can also affect Rev subcellular transport, so effects may be indirect.
3. Figure 5. legend states "competes" but there is no competition in this experiment. Check wording.

Version 1:

Reviewer comments:

Reviewer #2

(Remarks to the Author)

The authors have made a considerable effort to address the concerns raised during the initial review. A number of additional experiments have been included that support the overall conclusions of the manuscript. The manuscript was also significantly reorganized, as suggested, which improves the readability and logical flow of the manuscript.

A few very minor points could be noted:

Some figures could be better annotated to help the understanding of the figure. this pertains particularly to figure 3 but there are other examples throughout. For example, in figure 3, it would be useful to state on the y axis which type of RNA is being

measured.

Line 211, I think it should read 'Fig. 3d'.

Reviewer #3

(Remarks to the Author)

This is a revised version of a manuscript from Dorman et al. entitled "Nuclear retention of unspliced HIV-1 RNA as a novel reversible post-transcriptional block in latency". The paper studies regulation of the HIV-1 Rev protein and unspliced RNA nuclear export by host proteins Matrins 3 (MATR3) and MTR4; nuclear proteins involved in RNA maintenance and turnover.

The authors were responsive to the primary review and the revision is better organized and provides stronger evidence for MATR3 and MTR4 playing regulatory roles in HIV-1 RNA regulation, with MATR3 or MTR4 knockdown causing either reductions or increases in cytoplasmic unspliced viral RNA levels that are Rev-dependent but with more minor effects on viral spliced transcripts that are Rev-independent. Co-IP experiments coupled with cell imaging experiments support the notion that MATR3, MTR4, and Rev form complexes in the nucleus with unspliced RNA, and an interesting result shown in Figure 3g indicates that Rev may displace MTR4 from unspliced RNA, potentially describing a new way that Rev circumvents an antiviral host factor. The paper concludes with evidence for MATR3 upregulation in donor-derived HIV-infected T cells by PHA, and argues that a common feature of defective latency reversal tracks to nuclear retention of unspliced viral RNA.

Overall, this is an interesting study and there were strong efforts to improve organization and experiments addressing the proposed Rev/MATR3/MTR4/RNA interaction. Data using a drug that was proposed to target Rev (although may be a more general inhibitor of RNA metabolism), ABX464, were removed, that may render some of the results less controversial. That said, there are still significant gaps in the author's efforts to link Rev function and MATR3/MTR4 to latency reversal, summarized as follows:

1. Most importantly, it is not convincing from the data in this paper that nuclear retention of unspliced HIV-1 RNA is a major or novel form of latency. Regarding novelty, as noted by the authors post-transcriptional blocks have been suggested previously (refs 31-38) including unspliced RNA retention in the nucleus. If the argument for novelty here is that there is a link to MATR3/MTR4, Figure 4's data are inconsistent with this hypothesis considering that PHA has no effect on unspliced RNA export even though it upregulates MATR3. Only romi is shown to enhance cytoplasmic abundance (for a subset of donor cells) but the mechanism is not addressed (possible it downregulates MTR4?). It is also unusual in a latency study to see all donor cells exhibiting high levels of nuclear RNA, i.e. are they really reservoir cells if none of them are transcriptionally inactive? Second, shouldn't the short and long unspliced primer sets give similar results? they should both be components of the same Rev-dependent full-length transcript pool. Third, for the experiments in J-lats demonstrating that MATR3 helps during reactivation- this would not be unexpected for a Rev co-factor under conditions of either active replication or latency reversal- so I am convinced that MATR3 is a Rev co-factor but not sure if I see a strong link to latency reversal. I still think that a much stronger connection between latency reversal and MATR3/MTR4 is needed for this to be a rigorous study featuring data consistent with the authors' broad claims – need some data that truly support the idea of "...a block to Rev-dependent export that could result from differential levels of MATR3 and MTR4 and/or insufficient Rev levels" (Discussion lines 326-327).

2. In my opinion, the most compelling data are those that indicate that Rev either enhances or displaces MTR4 from unspliced RNA, and that this Rev attribute might be a "molecular switch" mechanism that overcomes a host restriction targeting unspliced RNA for degradation in the nucleus. If this could be confirmed in one or more orthogonal assays, it represent be a significant breakthrough in the context of our understanding of Rev function. However, more would need to be done here and it was not necessarily encouraging that differential effects were observed in Jurkats vs. primary cells for MTR4, and that Rev's effects on MATR3 were only addressed in primary T cells.

MINOR:

1. I found the authors' notion of "dynamic interplay" for the Rev/MTR4/MATR3/RNA complexes distracting and inaccurate. There are no kinetic analyses in the ppaer so I'm not sure I follow the argument, apparently based on co-localization and measurements of nuclear punctum volume.

2. Note: Figure 3C typo y-axis ("change")

3. Some paragraphs very long and could be broken up for improved readability.

ANSWERS TO THE REVIEWERS

Following is a point-by-point response to the reviewers' comments. Reviewer's comments are shown *in italic*. Our responses are in **blue**.

Reviewer #1 (Remarks to the Author):

The manuscript, "Nuclear retention of unspliced HIV-1 RNA as a novel reversible post-transcriptional block in latency" by Agnieszka Suder et al. attempts to characterize the activity of a small molecule Rev inhibitor ABX-464 in a contribution to HIV-1 latency. The authors use a J-Lat9.2 cell line harbouring an intact HIV-1 genome with multiply spliced HIV-1 RNAs detectable by GFP fluorescence. Using various manipulations including siRNA-mediated depletion, they implicate a relationship between related host factors MATR 3 & 4, a previously described phenomenon found to be important for nucleocytoplasmic export of RRE-containing RNAs. In general, the manuscript describes novelty as there is very little available information about such a post-transcriptional block in latent HIV-1 infections. Moreover, little is known about the involvement of the exosome in maintaining a pool of HIV-1 RNAs in latently infected cells.

This reviewer would offer a few required suggestions on how to convincingly show involvement of the MATR3 and 4 axis on RNA nuclear retention in latently infected cells.

1. The images shown in Supp Figure 2 are not convincing. Although T cells are generally difficult to image considering their large nuclei, the authors did not present control images. Moreover, a focal plane of many cells in culture should be presented for a more convincing presentation.

Thank you very much for your comment. We have included new images showing both HIV+ and HIV- cells in a focal plane containing multiple cells (new Supplementary Fig. 1c).

2. The authors employ qRT-PCR in many of the figures reflecting nuclear vs cytoplasmic preparations. Something is missing to instill confidence in these analyses. qRT-PCR is sensitive, but perhaps a bit too sensitive. Northern blotting could be performed to echo these data. The authors might expect the reader to take a leap of faith in accepting the data as written; a more convincing representation of the data is warranted for Nat. Comms.

We appreciate the reviewer's thoughtful comment. To address this concern, we would like to highlight that, in addition to our RT-qPCR data, we implemented an alternative approach—immuno-RNA FISH—to detect and quantify HIV RNA at the single-cell level. As illustrated in new Fig.1, our findings reveal a consistent trend between qPCR and RNA FISH results. By both approaches, we showed that depletion of MATR3 results in the downregulation of both nuclear and cytoplasmic Rev-dependent HIV-1 transcripts. Immuno-RNA FISH, like Northern blotting, provides relative abundance data of the RNA of interest, while RT-qPCR provides quantitative data. Moreover, immuno-RNA FISH assesses viral RNA at a single-cell level, while RT-qPCR (and Northern blot, in this matter) complements RNA FISH by quantifying the average relative RNA level within the entire sample. Additionally, we made an effort to quantify our RNA FISH signal by automatically counting the number of nuclear and cytoplasmic HIV-1 RNA spots and measuring their volumes.

We strongly believe that RT-qPCR and immuno-RNA FISH complement each other. As noted by the reviewer, Northern blot is less sensitive and less quantitative than RT-qPCR. We believe that the immuno-RNA FISH assay serves as a complementary method to RT-qPCR, providing an alternative approach to RNA quantification.

3. *The data with the 12 PLWH are quite skewed and are confusing to this reviewer. The authors should use these primary cells and examine the distributions of viral RNAs by imaging analyses.*

We used 16 PLWH for Fig. 6A-D (new Fig. 4b-e) and 6 PLWH for Fig. 6E-J (new Fig. 4f-k). Across all 22 PLWH, we observed nuclear enrichment of long US HIV-1 RNA with almost no detectable levels of cytoplasmic long US HIV-1 RNA. We believe our data convincingly demonstrate the nuclear retention of US HIV-1 RNA in patients. We utilized semi-nested qPCR, the most sensitive method available, to detect the limited amounts of viral RNA in unstimulated samples from ART-treated patients. HIV latently infected cells are extremely rare within a pool of CD4⁺ cells, making imaging analyses very challenging and insensitive since a cell must express a significant amount of RNA to be scored as positive. In contrast, the semi-nested qPCR used in this study is much more sensitive. The observed phenomenon of nuclear retention will be characterized in future studies and different imaging methods such as RNA FISH- flow to isolate the rare HIV⁺ cells will be evaluated.

4. *The siRNA knockdown western blot is not convincing. A mere two-lane blot demonstration as presented is rather elementary at this stage (Figure 1D).*

We agree that at this stage the western blot is rather too simplistic. We removed this result and instead we repeated the western blot to show MATR3 levels in nuclear and cytoplasmic fraction (new Fig. 1c).

5. *Legends are incomplete and unclear. E.g., legend text for Figure 4D is missing and the title of Figure 3 is missing something. This reviewer suggests to carefully edit the legends for completeness.*

Thank you very much for this comment. We revised the legends throughout the manuscript.

6. *Data presented in Figure 6 on nuclear retention of RNAs are interesting, but the authors should necessarily isolate relevant HIV-1-tropic cell types from pooled PBMCs for this analysis.*

Thank you for your comment. Most of the HIV reservoir resides in CD4⁺ cells, with CD4⁺ T cells being the primary target of HIV infection. There are indications that other than CD4⁺ cell types like MDM can also harbor HIV proviruses, as highlighted in a recent paper in Nature Microbiology (Veenhuis *et al.*, PMID: 36973419) making the HIV-1 reservoir even more complex. However, our focus in this study was to compare HIV-1 RNA levels within cells, specifically in the nucleus vs cytoplasm. Removing uninfected cells from PBMCs to isolate only the HIV⁺ cells would likely not alter the relative nuc/cyto ratios observed. In the future, we plan to assess the nuclear/cytoplasmic ratios in different CD4⁺ T subtypes - such as naïve, central memory (TCM), effector memory (ECM), regulatory (Treg) cells and in MDM to provide more detailed information on the observed nuclear retention phenomenon within different target cells. This technically demanding task to isolate minute fractions of different subtypes is beyond the scope of the current study.

7. *The Discussion is interesting too, but it reads a bit like a review and is not interpretative of the data shown in the manuscript.*

We thank the reviewer for his/her comments. We performed substantial reorganization of the discussion to better discuss the results and findings from the current study. Additionally, we removed general descriptions from the Discussion to the Introduction (lanes 90-93 and 100-104) to read less like a review indeed.

Reviewer #2 (Remarks to the Author):

Suder et al. have investigated post-transcriptional blocks to reactivation from latency of HIV-1. Indeed, novel strategies, including the targeting of post-integration, post-transcriptional processes, are valuable additions to anti-HIV therapy. In this study, they have focused on the export of unspliced HIV-1 RNAs, which is dependent of the viral protein Rev. They suggest a model in which Rev and a known partner, Matr3 (MATR3), favor the export of unspliced RNAs containing the rev-responsive element (RRE) in reactivated cells, whereas in latently infected cells in which neither MATR3 or Rev are highly expressed, the RNAs are bound by MTR4 helicase and targeted to the nuclear exosome for degradation. However, several aspects of the model have not been clearly demonstrated. This reviewer also feels that the organization of the manuscript is not optimal. The links between the different sections and how they relate to the overall model is not always clear.

1. To address the importance of Rev in viral reactivation, the authors have relied on the use of a drug, ABX464. While initially described as an inhibitor of Rev, it's mechanism of action remains unclear. It has not been shown to bind Rev, but instead binds the Cap-Binding-Complex. Even so, the mechanism by which ABX464 affects the export of unspliced HIV RNAs is fairly controversial. Recent studies clearly show that ABX464 upregulates miR124 in immune cells and induces a strong anti-inflammatory effect. The drug is being trialled for treatment of SARS-Cov2 infections for example. Furthermore, a recent study confirmed that ABX464, known clinically as Obefazimod, reduced the HIV reservoir, upregulated mir124 and reduced chronic inflammation in HIV patients. Thus, it seems unlikely that ABX464 is a specific inhibitor of Rev, and importantly for the present study, may very likely affect the immune activation of the cells, which is well known to be important for HIV-1 infection. Thus, it's difficult to interpret the results using this drug shown in Figures 1, 2 and 7, in terms of a specific effect on Rev.

We thank the reviewer for this important comment. We agree that ABX464 is likely not a specific inhibitor of Rev. The initial mechanism of action was characterized by Campos *et al.*, *Retrovirology* 2015 showing that ABX464 did not interfere directly with Rev but with Cap-binding complex (CBC) on unspliced HIV-1 RNA, preventing the export of Rev-dependent viral transcripts, without affecting spliced HIV-1 RNA as well as cellular splicing. The authors concluded that ABX464 targets Rev-mediated viral RNA biogenesis and interferes with Rev-mediated functions. In our study, we demonstrated an impact of ABX464 on Rev-dependent but not Rev-independent transcripts in a dose-dependent manner similarly to Campos *et al.* thus, we could distinguish the Rev-dependent and Rev-independent actions of ABX464. However, we agree with the reviewer that its specific mechanism of action is rather controversial because of induction of the splicing of lncRNA 0599-205 resulting in the generation of anti-inflammatory microRNA miR-122 which indeed brings complexity to the interpretation of some of our results. As such, we reorganized the manuscript by focusing less on ABX464. We moved the data from old Fig. 1 and Fig. 2 to supplementary file (new Suppl. Fig. 2) as ABX464 was used by us in this study more like a tool - not a principal focus in this work. As alternative to Rev inhibition is Rev depletion we therefore used a full-length molecular clone mutated in open reading frame of Rev from Luban's laboratory (McCauley *et al.*, DOI:10.1038/s41467-018-07753-2) and confirmed dependence of Rev in primary model of CD4⁺ T cells and in Jurkat cells. We also corrected in the new version of our manuscript that ABX464 inhibits function of Rev (lane 140) not Rev itself. We also removed ABX464 from our Discussion section as it is not the focus of our work.

2. The authors found that RRE-containing RNA in the nucleus is unstable in the absence of Rev partner, MATR3. They suggest an overall model in which MATR3 competes with MTR4, in which MATR3 promotes rev-dependent export or MTR4 promotes transcript degradation by targeting to nuclear exosome. However, several aspects of this model need to be confirmed. First, MATR3 and MTR4 would find to interact with HIV-1 RNA using an MS2 pull-down assay (Fig 4). This analysis should be complemented with standard RNA immunoprecipitation analysis, using antibodies to

endogenous proteins.

In the initial version of our manuscript, we performed RIP against endogenous MATR3, and we showed an increased interaction with HIV-1 RNA in siMTR4-treated cells. As asked by the reviewer, we complemented this result by performing additional RIPs against endogenous MATR3 and MTR4 in either primary model of CD4⁺ T cells or Jurkat cells infected with full-length molecular clones mutated or not in open reading frame of Rev (new Fig. 3). We demonstrated that Rev presence alters the interactions of either MATR3 or MTR4 with RRE-containing HIV-1 RNA, whose exhibit opposing HIV-1 RNA binding phenotypes.

3. Co-immunoprecipitation analysis showed that MATR3 interacts with MTR4 and with Rev, and that MTR4 interacts with MATR3 and with Rev. The authors conclude that all 3 proteins form a complex. However, these analyses do not show that all 3 factors are found in the same complex. The proteins could be in separate complexes (for example MATR3 and MTR4 and MATR3 and Rev), especially as the MATR3-MTR4 interaction appears quite weak, whereas MTR4 seems to interact very well with Rev. The authors should perform re-IPs to address whether all 3 proteins are found in the same complex. Indeed, according to the final model, all 3 proteins would not be expected to be found together on HIV RNA. Additionally, it is important to address whether these interactions are dependent on RNA by performing the IPs in the presence and absence of RNase. They next addressed the co-localization of MTR4, MATR3 and HIV-1 RNA by immune RNA-FISH. The results are not compelling. There does not seem to be a particularly significant overlap of the 3 molecules. However, it would be necessary to perform quantification and statistical analysis to test this. Again, given that the overall model suggests that these factors may compete for the RNA, would an overlap be expected?

We thank the reviewer for these very important comments that asks bottom-line questions regarding the MATR3-MTR4-Rev axis. To answer these, we performed new sets of IP experiments against MATR3 and against MTR4 coupled with nuclease treatment to address the nature of the interactions. We showed that MATR3 interacts with MTR4 *via* protein-protein while Rev interacts with the two *via* protein-RNA (new Fig. 2b-c). To further corroborate these findings, we performed re-IP experiment against MATR3 followed by IP against Rev and we showed that MATR3, MTR4, Rev are present in the same complex (new Fig. 2c). Moreover, as requested, we quantified the colocalization data between HIV-1 RNA and MATR3, HIV-1 RNA and MTR4, MATR3 and MTR4, as well as the triple colocalization of HIV-1 RNA, MATR3, and MTR4. This was achieved by measuring the number of overlapping spots and their respective volumes from 3D images (as presented in new Fig. 2f). As explained in lines 176-181: “Quantification of HIV-1 RNA-MATR3, HIV-1 RNA-MTR4, and MATR3-MTR4 colocalization revealed a 30%, 65%, and 18% overlap of spots, respectively, while their volume overlap was 41%, 43%, and 34%, respectively (Fig. 2f). Next, HIV-1 RNA-MATR3-MTR4 triple colocalization quantification showed 19% overlap between the spots and 39% overlap between their volume (Fig. 2f) supporting the idea of a dynamic and transient interaction between MATR3 and MTR4 with viral RNA”. Overall, our experimental data support a notion that all three proteins are found in the same ribonucleoprotein complex that is most likely highly dynamic, supporting the idea that these proteins may interact in a flexible, regulated manner during RNA processing.

4. In Figure 5, the authors depleted MTR4 and found that the abundance of Rev-dependent transcripts increased, and more transcripts were associated with MATR3, leading to the notion that MTR4 and MATR3 exhibit antagonizing activities. An alternative explanation could be that loss of MTR4 leads to accumulation of HIV RNA, and over-expression of Rev, together with its partner MATR3, allows the transcript to be exported to the cytoplasm where it is protected from degradation. In this case, the factors are not necessarily antagonistic, but rather Rev/MATR3 exports the RNA that has been

stabilized by loss of MTR4. If the authors wish to claim antagonistic activities of MTR4 and MATR3, additional experiments that directly address the antagonism must be provided. These sections (figures 4 and 5) could be reorganized to provide a clearer idea of how the authors interpret their data. In figure 4, they claim that the factors form a complex, but in figure 5, they claim they have antagonistic activities. The authors should reorganize or better explain the interpretation of the data to provide a clearer flow of their logic for the reader.

We thank the reviewer for these comments. As mentioned above, our new sets of experimental data (IP-/benzonase and reRIP) support that MATR3, MTR4 and Rev are found in the same ribonucleoprotein complex (new Fig. 2b-d). Our subsequent RIP experiments demonstrate opposite relationship between MATR3-HIV-1 RNA and MTR4-HIV-1 RNA that is dependent on Rev (new Fig. 2H, new Fig. 3). Opposing activities of proteins in a complex suggest that they may act as “molecular switches”. Indeed, there are several examples of “molecular switches” involved in tightly regulated cellular processes, such as signal transduction, cell cycle control, and gene expression. Additionally, recent study of Wang *et al*, *Genes Dev* 2019 (doi: [10.1101/gad.322602.118](https://doi.org/10.1101/gad.322602.118)) demonstrated a protein-protein interaction between MTR4 and NRDE2 with opposing activities to control the stability of mRNA. NRDE2 was demonstrated to negatively modulate the activity of MTR4 (by locking it in inactive state) to assure mRNA stability and export. Nature of this opposing interactions between MATR3 and MTR4 that likely create a dynamic balance to regulate the viral RNA is beyond the scope of this study and will be investigated in detail in the future through sets of experiments such as pull-downs of various MATR3 and MTR4 mutants, or by investigating the potential PT modifications of proteins that could act as “switches” in the presence or absence of Rev *etc*. Understanding how this balance is regulated could provide insights into the mechanisms of RNA stability and degradation. We discussed the potential role of MATR3-MTR4 acting as “molecular switches” in Discussion section (lanes 296-301).

As requested, we revised the text in the Results section to improve clarity and ensure a more coherent flow. To achieve this, we relocated the data related to ABX464 (previously in Figures 1 and 2) to new Supplementary Figure 2, as ABX464 was not a central focus of our study. Furthermore, we merged the data from the previous Figures 4 and 5 into a new Figure 1 to concentrate immediately on the role of MATR3. Additionally, we introduced a new Figure 3, which highlights the Rev-dependence of MATR3 and MTR4 binding to viral RNA, providing a more streamlined and focused presentation of our findings.

5. The data shown in Figure 6 is problematic for several reasons. In the experiments using 6 patients, the analysis sometimes ends up using 2 or 3 samples, which are not always the same set analysed between the various treatment conditions, and which are quite variable amongst themselves. There is no statistical analysis shown and it may even be quite difficult to perform statistics on these results.

Thank you for your insightful comments. Indeed, although we used cells from 6 PWH for the reactivation experiments, panels in Figure 6 (new Fig. 4) where we show percentages of HIV-1 RNA in cytoplasm sometimes contain less than 6 participants per condition. The reason for that is that if HIV-1 RNA in both nucleus and cytoplasm was undetectable, then the percentage cannot obviously be calculated and is not shown (as explained in the figure legend, lanes: 787-789). If the reviewer refers to the open circles (undetectables), then we would like to point out that undetectable values are censored to the detection limit of the assay and are always considered (lanes: 781-787).

6. Furthermore, what is the short unspliced RNA in the cytoplasm?

We understand the importance of clarifying the distinction between short and long unspliced (US) RNAs. In our manuscript, we addressed this in lines 244-248, where we explain that two sets of

primers were used to detect shorter (amplicon spanning the packaging signal and the beginning of gag ORF, minimal transcript length 342 nt) and longer (amplicon located in the middle of gag ORF, minimal transcript length 1045 nt) US HIV-1 transcripts, which we refer to as US-short and US-long, respectively. If the reviewer is inquiring about the presence of short unspliced HIV-1 RNA in the cytoplasm, particularly regarding how such short transcripts, lacking the Rev Response Element (RRE), could be exported, we would like to refer to our discussion in lines 334-341. Here, we propose that “Length of HIV-1 US transcripts seems also to be important for nuclear export, as we did not observe any nuclear retention of short US transcripts, in contrast to long US transcripts. It is possible that short transcripts are exported to the cytoplasm in an alternative, Rev-independent, fashion. Further studies are needed to understand the mechanisms involved and the role of such short viral RNA in latently infected PWH cells. Interestingly, a recent study from the Luban laboratory demonstrated that intron-containing RNA from the HIV-1 provirus activates innate immune signaling pathways⁴⁷. In future studies, it will be important to investigate whether the length of such transcripts determines inflammation.”

7. Also, it isn't clear what the conclusion of these experiments really is, and how it relates to the previous figures. This is another example of a suboptimal organization of the manuscript.

We thank the reviewer for this comment. We have reorganized the results from the original Figures 6 and 7 in a new Figure 4 to better connect them with the previous results, as the linkage was indeed not well explained. Now we start the paragraph on nuclear retention by saying (lines 228-238): “The opposing nature of the interaction between MATR3 and MTR4 with viral RNA triggered by Rev either stabilizing or degrading viral transcripts may shed light on post-transcriptional mechanisms that could be crucial for maintaining viral latency or enabling reactivation. The varying levels of these two factors in unstimulated versus activated cells may shift the balance, favoring one of these scenarios. Previously, we showed limiting levels of MATR3 in CD8+-depleted PBMCs from ART-treated PWH that correlated with limited ability of some LRAs to reactivate the virus. We therefore evaluated the levels of MATR3 and MTR4 in ex vivo cultures of PBMCs isolated from 3 healthy donors that were either unstimulated or activated with PHA. As previously published, MATR3 levels were limited in unstimulated cells and upregulated upon PHA treatment (Fig. 4A)⁴¹. Notably, MTR4 levels were abundant in both conditions (Fig. 4a). These varying levels of both factors in unstimulated cells suggests that the export of the viral RNA might be impaired.”

We have reorganized the results to first present the levels of MATR3 and MTR4 in unstimulated versus activated PBMCs, confirming previous findings that MATR3 levels are limited in unstimulated conditions, but MTR4 levels are abundant (new Fig. 4a). Next, we showed the nuclear and cytoplasmic HIV-1 RNA levels in patients' cells, which demonstrated the nuclear retention of unspliced HIV-1 RNA in 22 patients as the first evidence of a post-transcriptional reversible block (new Fig. 4b-k). At the end we propose a working model of viral RNA regulation in unstimulated versus stimulated HIV⁺ cells (new Fig. 4l). We slightly modified the model by placing MATR3 next to MTR4 to reflect better their protein-protein interactions. “Sorting” was removed from the initial model and “the molecular switch” was discussed instead in the Discussion section.

8. In fig 7, the use of ABX464 to demonstrate that MTR4 hampers viral reactivation via Rev complicates the interpretation of the experiment. In the presence of this anti-inflammatory drug, a higher proportion of HIV-1 transcripts may be aberrant, which would explain the increased association with MTR4. The authors should use an alternative strategy to demonstrate a dependence on Rev.

We agree with the reviewer that using ABX464 bring complexity to the interpretation of our results as also elaborated in point 1. We removed these results (old Fig. 7b-c) from the manuscript and instead we performed experiment in primary model of CD4+ T cells or Jurkat cells infected with full

length molecular clone mutated in open reading frame of Rev (new Fig. 3). With that alternative strategy we confirmed Rev dependence in determining the MATR3-MTR4 binding to unspliced HIV-1 RNA using more physiological models of HIV-1 infection.

Additional comments:

9. Many of the western blots are of poor quality. It would be preferable to have better quality blots to increase the robustness of the data.

We performed new sets of data and replaced old blots with new panels (new Fig. 1C and new Fig. 2B, C, and D)

10. Additional controls for nuclear and cytoplasmic fractions should be provided. For RNA analysis, it is more appropriate to quantify a nuclear RNA, such as 7SK or an snRNA. Quantifying DNA is not sufficient to assess contamination of the cytoplasmic fraction by the nucleoplasm. For immunoblotting analysis, a nucleoplasmic factor, such as a transcription factor or a splicing factor, should be included. Histones mark the chromatin fraction.

We thank the reviewer for these important comments. In this study we used our previously published biochemical fractionation protocol (Kula et al, *Retrovirology* 2011 and Kula et al, *Virology* 2013). For this study, we have initially performed sets of experiments to optimize the protocol for lymphocytic cells. We though evaluated increasing concentrations of NP-40 (0.1, 0.25 and 0.5%), and we assessed the purity of the nuclear and cytoplasmic fractions by measuring gapdh pre-mRNA levels by PCR (panel A). Moreover, we saw >7 Ct difference between nuclear (nuc) and cytoplasmic (cyto) fractions when gapdh pre-mRNA was measured by RT-qPCR (panel B). As shown in panel B, we observed the biggest difference for 0.5% NP-40 (in cyto, gapdh pre-mRNA levels were undetectable). As such, we selected 0.5% of NP-40 in our fractionation protocol for further experiments. On the other hand, to assure the purity of nuclear fraction, we searched for “cytoplasmic-only” RNA. We checked the levels of CoxII mitochondrial RNA (panel A). However, due to existing pseudogenes of this gene in a genome (doi:10.1101/gr.227202), the levels were largely similar between both fractions (panels A-B). Of note, we saw the same pattern for CoxII mtRNA in patients’ samples (data can be revealed upon reviewer’s request). That is why we relied on Western blot results overall by analyzing the cytoplasmic and nuclear markers. As requested by the reviewer, we added another nuclear marker to assess the purity by Western blot using antibodies against splicing factor PSF and we confirmed that our cytoplasmic fractions are not contaminated by nucleoplasm (new Fig 1c). Unfortunately, we could not detect neither PSF nor other nuclear proteins (such as METTL3, ALKBH5, SC-35) in patient samples. This is most likely due to their liming levels in unstimulated cells as we showed previously for MATR3 (Sarracino et al, *mBio* 2018, doi: 10.1128/mBio.02158-18).

B

SAMPLE	TARGET	Ct Mean	Ct SD	Ct _{cyto} -Ct _{nuc}
0.5 nuc	pre mrna gapdh	28,70	0,32	
0.5 cyto				
0.25 nuc		26,60	0,06	
0.25 cyto		36,83	0,18	10,23
0.1 nuc		27,13	0,08	
0.1 cyto		34,95	0,49	7,81
0.5 nuc	mrna gapdh	16,98	0,08	
0.5 cyto		20,04	0,10	3,07
0.25 nuc		16,13	0,02	
0.25 cyto		18,89	0,08	2,76
0.1 nuc		16,35	0,20	
0.1 cyto		17,94	0,02	1,59
0.5 nuc	cox ii	21,49	0,07	
0.5 cyto		16,15	0,12	5,33
0.25 nuc		20,37	0,10	
0.25 cyto		15,21	0,01	5,17
0.1 nuc		21,84	0,08	
0.1 cyto		14,64	0,26	7,20

Reviewer #3 (Remarks to the Author):

In this study, Suder et al. study blocks to HIV RNA nuclear export in the context of viral latency reversal, focused largely on the effects of the drug ABX-464 that is known to reduce Rev activity and a host factor, Matr3, implicated as a positive regulator of Rev.

To me, this is interesting work but it feels like two developing stories. In the first part of the paper (Figures 1 and 2) the authors show that ABX-464 or MATR3 shRNA each reduce virus output during attempts to reverse HIV latency in a JLAT 9.2 model. However, based on the literature blocking Rev should block viral output in any setting so that the significance of these results is not very clear. Later (Figure 7), MATR3 levels were shown to be upregulated by LRAs in human donor cells (Figure 7A) but the authors did not attempt to correlate these effects (or, alternatively, whatever ABX-464 is targeting) to their patient cell post-transcriptional block.

The second part of the paper (Figures 3-5) is focused more on mechanism and addresses the role of MATR3 in regulating HIV RNA stability, suggesting a link between MATR3 and MTR4 (a regulator of the RNA exosome). I think the data are convincing that MATR3 regulates HIV RNA stability, and I also found it interesting that MATR3 looks to differ from ABX-464 in its overall mode of action. However, the argument for a direct link to MTR4 in this pathway looks premature, considering that MTR4 depletion appears to play a general role in regulating viral RNA levels whether or not they are Rev-dependent (Figure 5).

We made substantial changes in the revised manuscript to better link the different parts to make one logic story about dynamic interplay between MATR3 and MTR4 on viral RNA that is dependent on Rev. With new sets of data, we show that Rev determines the binding of either MATR3 or MTR4

to HIV-1 RNA in primary model of HIV-1 infection with full-length molecular clone mutated in open reading frame of Rev. More specifically, we demonstrated by RIP approach that lack of Rev results in increasing binding of unsliced HIV-1 RNA to MTR4 (new Fig. 3h). As highlighted in the Discussion, we acknowledged that MTR4 has reported transcriptional role in HIV-1 transcription repression (lanes 294-295). In addition to transcriptional role, our data demonstrate that MTR4 play also a post-transcriptional role as reflected by our IP and re-IP experiments showing the MTR4 interaction with Rev and by RIPs in infection with full-length molecular clone mutated in open reading frame for Rev (discussed in lines 295-300).

Overall, this is a broad study with interesting observations here but a weakness in cohesion. Some more specific comments are as follows.

Major:

1. Figure 2. In 2D it seems strange that a >1000-fold increase in RNA levels based on qRT-PCR doesn't translate to the FISH assay where differences are quite modest. Also effects in Figures 2 and 3 are minor compared to Figure 1. Are these assays comparable? What explains these discrepancies?

While we see a general trend that is similar between qPCR and RNA FISH data, we indeed do not see the same fold changes. RNA FISH gives information about localization and relative abundance of RNA of interest in a single cell while RT-qPCR complements RNA FISH by quantifying the average relative level of RNA within the entire sample. We made an effort to quantify our RNA FISH signal by automatically counting the number of HIV^{gag} RNA spots and measuring their volumes however, this approach does not allow absolute RNA quantification as it does not achieve single-molecule sensitivity and resolution. HIV-1 RNA spots/condensates most likely contain multiple RNA molecules which we cannot detect, moreover, the compactness of spots may hamper probes binding as well.

2. Related, in 2E, what is the resolution of the system that allows the authors to determine that dot volumes are greater or less than 1 μ m? Presumably the argument here is that these are not single RNAs but granules of some nature, correct? The rigor of this system and relevant caveats could be better explained. Related, in 3E if MATR3 is affecting stability then shouldn't we would expect a bigger effect on volume in this setting relative to the drug?

We apologize for the lack of detailed information regarding the resolution of our system. We have added the important information regarding the objective used and its numerical aperture (1.46) which is high enough to distinguish between object that are smaller or larger than 1 μ m³. For dot volume measurements, we used segmentation and quantitative image analysis software to assign a volume to each detected signal (dot) and objects higher than 5 voxels (which corresponds to 0.0008523 μ m³) were considered as now described in the Materials and Methods section (lanes: 468-470). Our analysis did not focus on quantifying individual RNA molecules, as this would require calibration with specialized beads. Instead, the detected dots likely represent clusters or aggregates, which may correspond to RNA granules. Regarding the question about the old Fig. 3E showing MATR3 impact on HIV-1 RNA spots volume and comparing it with ABX464's impact, we now agree that the mode of action of ABX464 is somewhat controversial, as raised by reviewer 2 and elaborated by us in points 1 and 8 of the reviewer's comments. Given the complexity of ABX464's mechanisms, it is challenging to interpret the data obtained with this compound and directly compare it to the MATR3 phenotype. We acknowledge the difficulties in drawing clear parallels between the two, and further investigation would be necessary to clarify these interactions.

3. Figure 3H is hard to interpret without a TNF alone control

As use of ABX-464 was of concern to the reviewer 2, we removed this result and instead we performed experiment in primary model of CD4+ T cells or Jurkat cells infected with full length molecular clone mutated in open reading frame of Rev (new Fig. 3). With that alternative strategy we confirmed Rev dependence in determining the MATR3-MTR4 binding to unspliced HIV-1 RNA.

4. *Figure 4. There are some confusing discrepancies in these experiments, e.g., 4A tracks Tat and RNA, while 4B and 4C track Rev but not Tat or the viral RNA. Also, wouldn't the most relevant experiment to look at MTR4/exosome-HIV interactions with or without MATR3?*

We apologize for the confusion. Tat, which is a viral transcriptional activator was used to activate HIV-1 transcription from the viral vector in our co-transfection experiment. Tat as it is known to bind TAR on viral RNA was used as our internal control. Next, as Rev is known to bind to MATR3 (Kula et al, *Retrovirology* 2011), Rev was used as control in immunoprecipitation against MATR3. We explained now in the new version of the manuscript the rationale behind tracking Tat in the new Fig. 2a (lanes 155-157) and Rev in the new Fig. 2b-d (lanes 159-160).

5. *Figure 5. why move to 293T cells for these experiments?*

6. *Figure 7B. Why move to U1 cells for these experiments?*

We acknowledge the reviewer's concerns regarding the use of different cell lines. Experiment shown in old Fig. 7b was removed due to reviewers' concerns reading specificity of ABX464. Next, we confirmed the importance of MATR3-MTR4-Rev regulatory axis on viral RNA in more physiological cellular setting of primary CD4⁺ T and Jurkat CD4⁺ T cells infected with full length HIV-1 molecular clones mutated in open reading frame of Rev (new Fig. 3).

Minor:

1. *Figure 3. "Retain" may be wrong word choice if it's stability.*

The new title for Fig. 3 (new Fig. 1) is "MATR 3 stabilizes Rev-dependent HIV-1 RNA during reactivation from latency".

2. *Figure 3F. Should be noted here that ActD can also affect Rev subcellular transport, so effects may be indirect.*

We would like to clarify that in our study, Actinomycin D (ActD) was used, which inhibits transcription but does not directly affect the subcellular transport of Rev. The confusion might stem from Leptomycin B, which is known to inhibit Rev export by blocking CRM1. We hope this clarification resolves any misunderstanding.

3. *Figure 5. legend states "competes" but there is no competition in this experiment. Check wording.*

Reviewer is right, we do not provide evidence of direct competition between MATR3 and MTR4 while we show the opposing activities on viral RNA which we now explained better in the Results and Discussion. Fig. 5 is now part of the new Fig. 2. The new title is: "MATR3/MTR4/Rev ribonucleoprotein complex regulates the fate of Rev-dependent HIV-1 RNA".

ANSWERS TO THE REVIEWERS

Following is a point-by-point response to the reviewers' comments. Reviewer's comments are shown *in italic*. Our responses are in **blue**.

Reviewer #2 (Remarks to the Author):

The authors have made a considerable effort to address the concerns raised during the initial review. A number of additional experiments have been included that support the overall conclusions of the manuscript. The manuscript was also significantly reorganized, as suggested, which improves the readability and logical flow of the manuscript.

We thank the reviewer for recognizing our efforts to thoroughly address the concerns raised.

A few very minor points could be noted:

Some figures could be better annotated to help the understanding of the figure. this pertains particularly to figure 3 but there are other examples throughout. For example, in figure 3, it would be useful to state on the y axis which type of RNA is being measured.

We annotated Figure 3 to help the readability.

Line 211, I think it should read 'Fig. 3d'.

We corrected the mistake.

Reviewer #3 (Remarks to the Author):

This is a revised version of a manuscript from Dorman et al. entitled "Nuclear retention of unspliced HIV-1 RNA as a novel reversible post-transcriptional block in latency". The paper studies regulation of the HIV-1 Rev protein and unspliced RNA nuclear export by host proteins Matr3 (MATR3) and MTR4; nuclear proteins involved in RNA maintenance and turnover.

The authors were responsive to the primary review and the revision is better organized and provides stronger evidence for MATR3 and MTR4 playing regulatory roles in HIV-1 RNA regulation, with MATR3 or MTR4 knockdown causing either reductions or increases in cytoplasmic unspliced viral RNA levels that are Rev-dependent but with more minor effects on viral spliced transcripts that are Rev-independent. Co-IP experiments coupled with cell imaging experiments support the notion that MATR3, MTR4, and Rev form complexes in the nucleus with unspliced RNA, and an interesting result shown in Figure 3g indicates that Rev may displace MTR4 from unspliced RNA, potentially describing a new way that Rev circumvents an antiviral host factor. The paper concludes with evidence for MATR3 upregulation in donor-derived HIV-infected T cells by PHA, and argues that a common feature of defective latency reversal tracks to nuclear retention of unspliced viral RNA.

Overall, this is an interesting study and there were strong efforts to improve organization and experiments addressing the proposed Rev/MATR3/MTR4/RNA interaction. Data using a drug that was proposed to target Rev (although may be a more general inhibitor of RNA

metabolism), ABX464, were removed, that may render some of the results less controversial. That said, there are still significant gaps in the author's efforts to link Rev function and MATR4/MTR4 to latency reversal, summarized as follows:

1. Most importantly, it is not convincing from the data in this paper that nuclear retention of unspliced HIV-1 RNA is a major or novel form of latency. Regarding novelty, as noted by the authors post-transcriptional blocks have been suggested previously (refs 31-38) including unspliced RNA retention in the nucleus.

We would like to clarify that the message of our work is not that the nuclear retention is a novel latency mechanism, as we (Sarracino et al, 2018) and others (Lassen et al, 2006) previously proposed but rather a first study dwelling into the mechanism of this kind of latency. Through our work, we provide new post-transcriptional mechanistic insights into the fate of unspliced HIV-1 RNA during latency and reactivation/infection (interplay with RNA degradation via MTR4 and stabilization via MATR3 with Rev being a central switch) that may guide future research towards strategies focusing on post-transcriptional mechanisms to either maintain viral latency or enable reactivation in the context of “block-and-lock” and “shock-and-kill” cure strategies, respectively. We amended the title of our manuscript to: “Nuclear Retention of Unspliced HIV-1 RNA as a Reversible Post-Transcriptional Block in Latency”.

If the argument for novelty here is that there is a link to MATR3/MTR4, Figure 4's data are inconsistent with this hypothesis considering that that PHA has no effect on unspliced RNA export even though it upregulates MATR3. Only romi is shown to enhance cytoplasmic abundance (for a subset of donor cells) but the mechanism is not addressed (possible it downregulates MTR4?).

The lack of PHA in reversing the block could be due to the overall weak viral reactivation effect we observed, potentially driven by cytotoxicity of PHA. Romidepsin, a potent LRA, indeed increased cytoplasmic viral RNA abundance, demonstrating that the block could be reversed. While it would have been interesting to assess whether romidepsin downregulates MTR4, as suggested by the reviewer, here we exclusively aimed to explore if the block can be reversed using different LRAs. Moreover, given that romidepsin is an epigenetic drug, its mechanism of action is likely more complex than solely reducing MTR4 levels.

It is also unusual in a latency study to see all donor cells exhibiting high levels of nuclear RNA, i.e. are they really reservoir cells if none of them are transcriptionally inactive?

We would like to emphasize that the data in the Figure 4 represent bulk HIV RNA levels in study participants, not individual cells. Therefore, these data do not allow inference of the frequencies of transcriptionally silent versus active infected cells. Notably, previous studies have shown that a subset of reservoir cells in most PWH on ART do express cell-associated unspliced HIV RNA (e.g., Yukl et al., 2018) as also mentioned in the Introduction section (lines: 92-94).

Second, shouldn't the short and long unspliced primer sets give similar results? they should both be components of the same Rev-dependent full-length transcript pool.

Studies from Yukl laboratory demonstrated that there are very few full-length unspliced transcripts in PWH on ART due to transcription elongation blocks resulting in a gradient of unspliced HIV transcript abundance. Therefore, with the "short" and "long" primer sets we do measure different pools of transcripts because a number of transcripts will terminate between the target sites of these primer sets. For better clarity we included the following descriptions in the Results section (lines 249-250): “Because of possible latency blocks to HIV-1 transcription elongation resulting in gradient of unspliced HIV-1 transcript abundance³¹...” and in the Discussion (lines 344-346): “Indeed, as demonstrated by Yukl laboratory, a gradient of unspliced HIV-1 transcript abundance is observed in CD4+ T cells from PWH on ART due to the constant termination of transcription during elongation (31,32).”

Third, for the experiments in J-lats demonstrating that MATR3 helps during reactivation- this would not be unexpected for a Rev co-factor under conditions of either active replication or latency reversal- so I am convinced that MATR3 is a Rev co-factor but not sure if I see a strong link to latency reversal. I still think that a much stronger connection between latency reversal and MATR3/MTR4 is needed for this to be a rigorous study featuring data consistent with the authors' broad claims – need some data that truly support the idea of “..a block to Rev-dependent export that could result from differential levels of MATR3 and MTR4 and/or insufficient Rev levels” (Discussion lines 326-327).

While additional studies could definitely further refine our model, we believe our data already provide strong mechanistic insights into how MATR3/MTR4/Rev regulate the fate of viral RNA during latency and reactivation/infection using well established *in vitro* model of latency (J-Lat 9.2), co-transfection experiments to define the MATR3-MTR4-Rev complex and primary CD4+ T cells model of infection using full-length and Rev-mutated HIV molecular clones. These findings provide new insights into how host RNA-binding proteins modulate viral RNA export especially for the interplay with RNA degradation versus stabilization/retention mechanisms with Rev being a central switch. Regarding the direct connection between MATR3/MTR4 levels and Rev-dependent export during reactivation, we acknowledged in the Discussion section the limitation of our study in lines 359-363: “A limitation of our study is that, although we were able to demonstrate evidence for the roles of MATR3 and MTR4 in the posttranscriptional regulation of US HIV-1 RNA in co-transfection experiments, in *in vitro* J-Lat 9.2 model of latency and in primary cell model of HIV-1 infection, their direct roles in post-transcriptional latency and reactivation in primary cell models of latency or PWH cells remain to be established.”

2. In my opinion, the most compelling data are those that indicate that Rev either enhances or displaces MTR4 from unspliced RNA, and that this Rev attribute might be a “molecular switch” mechanism that overcomes a host restriction targeting unspliced RNA for degradation in the nucleus. If this could be confirmed in one or more orthogonal assays, it represent be a significant breakthrough in the context of our understanding of Rev function.

Understanding the “switch” is beyond the scope of this study, as it would require extensive functional assays and binding experiments with different MATR3 and MTR4 mutants in the presence and absence of Rev. However, a recent study has suggested a potential regulatory mechanism. Indeed, Wang et al. has demonstrated that MTR4 interacts with NRDE2, which locks MTR4 in a closed conformation, inhibiting its activity

and promoting mRNA stability and export (as mentioned in Discussion section, lines 307-310). During the revision process, we further explored whether the proposed switch involved NRDE2. Using RIP in a primary CD4⁺ T cell model—similar to our approach with MTR4 in Figure 3e—we assessed viral RNA binding to NRDE2. Surprisingly, we observed a significant increase in viral RNA binding to NRDE2 in cells infected with a Rev-deleted molecular clone. This was unexpected given NRDE2's known role in negatively regulating MTR4, further highlighting the complexity of viral RNA fate regulation and the need for further future investigation.

However, more would need to be done here and it was not necessarily encouraging that differential effects were observed in Jurkats vs. primary cells for MTR4, and that Rev's effects on MATR3 were only addressed in primary T cells.

We respectfully argue that primary T cells more accurately reflect the physiological environment of viral infection compared to transformed cell lines such as Jurkat-based in vitro model of latency (J-Lat 9.2). That is why we did not address the effects of MATR3 in the J-Lat 9.2 model as we did for MTR4, also because we have already addressed the MATR3 impacts in Fig. 1 and 2. Regarding MTR4's lack of effect in primary cells and as explained by us in the Results section lines 226-229:

“The pronounced MTR4 phenotype in Jurkat cells over primary CD4⁺ T cells highlights the complex and heterogenous nature of molecular mechanisms that contribute to the HIV-1 gene expression in primary cells.”

MINOR:

1. I found the authors' notion of “dynamic interplay” for the Rev/MTR4/MATR3/RNA complexes distracting and inaccurate. There are no kinetic analyses in the ppaer so I'm not sure I follow the argument, apparently based on co-localization and measurements of nuclear punctum volume.

We interpret the partial overlap in MATR3-MTR4-HIV RNA colocalization (Fig. 2h) as a possible reflection of the dynamic nature of these interactions. However, we acknowledge the reviewer's concern regarding the term "dynamic interplay," as our study does not include kinetic analyses. To address this concern, we revised the text by removing "dynamic" while retaining "interplay" when describing the observed colocalizations among Rev, MTR4, MATR3, and RNA.

2. Note: Figure 3C typo y-axis (“change”)

Corrected.

3. Some paragraphs very long and could be broken up for improved readability.

We broke up some long paragraphs in Discussion section to improve readability.